# Model Steering: Learning with a Reference Model Improves Generalization Bounds and Scaling Laws

Xiyuan Wei [1]  Ming Lin [2]  Fanjiang Ye [3]  Fengguang Song [3]  Liangliang Cao [4]  My T. Thai [5]  Tianbao Yang [1]

## Abstract

This paper formalizes an emerging learning paradigm that uses a trained model as a reference to guide and enhance the training of a target model through strategic data selection or weighting, named **model steering**. While ad-hoc methods have been used in various contexts, including the training of large foundation models, its underlying principles remain insufficiently understood, leading to sub-optimal performance. In this work, we propose a theory-driven framework for model steering called **DRRho risk minimization**, which is rooted in Distributionally Robust Optimization (DRO). Through a generalization analysis, we provide theoretical insights into why this approach improves generalization and data efficiency compared to training without a reference model. To the best of our knowledge, this is the first time such theoretical insights are provided for the new learning paradigm, which significantly enhance our understanding and practice of model steering. Building on these insights and the connection between contrastive learning and DRO, we introduce a novel method for Contrastive Language-Image Pretraining (CLIP) with a reference model, termed DRRho-CLIP. Extensive experiments validate the theoretical insights, reveal a superior scaling law compared to CLIP without a reference model, and demonstrate its strength over existing heuristic approaches. Code is released at github.com/Optimization-AI/DRRho-CLIP

## 1. Introduction

With the success of large foundation models, numerous companies and research groups have entered the race to develop

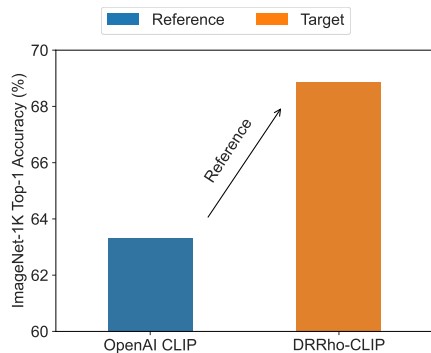

Figure 1: Comparison between a target model (ViT-B/16) trained by the proposed DRRho-CLIP and the reference model it leverages. OpenAI CLIP (ViT-B/32) was trained on a private 400M dataset with 12.8B samples seen and 32768 batch size. DRRho-CLIP model was trained on DFN-192M with 1.28B samples seen and 5120 batch size, and using OpenAI CLIP as a reference model [1].

state-of-the-art models. While the data and code are often proprietary, the resulting models are sometimes released publicly, such as the CLIP models from OpenAI (Radford et al., 2021) and the Llama models from Meta (Dubey et al., 2024). This raises an intriguing question:

"*How can we leverage public models to improve training of a target model on custom datasets?*"

To study this question, we explore an emerging learning paradigm that leverages a trained model as a reference to guide and enhance the training through strategic data selection or weighting. We refer to this paradigm as **model steering**. Unlike the widely adopted knowledge distillation framework, model steering does not assume that the reference model is a stronger teacher; in fact, it can lead to the training of a model that ultimately surpasses the reference model in performance, i.e., enabling weak to strong generalization (cf. Figure 1).

A few works have studied learning with a reference model (LEAR) in different contexts and demonstrated its effective-

---

[1]Texas A&M University [2]Oracle [3]Indiana University [4]Google [5]University of Florida. Correspondence to: Tianbao Yang <tianbao-yang@tamu.edu>.

*Proceedings of the 42nd International Conference on Machine Learning*, Vancouver, Canada. PMLR 267, 2025. Copyright 2025 by the author(s).

[1]Our training took 376 GPU hours on 8 H100 (2 days), OpenAI CLIP (ViT-L/14) model was trained on 256 V100 with 12 days, which gives an estimate of 256*12*24/11.6=6356 GPU hours for training ViT-B/32 as its FLOPs is 11.6 smaller.

ness in accelerating the training of various models, including classification models (Mindermann et al., 2022; Evans et al., 2025), large language models (Lin et al., 2024; Xie et al., 2023), CLIP models (Evans et al., 2024a;b). An interesting idea emerging from these studies is to perform online data selection, sampling, or data weighting using the reference model through a shifted loss called the RHO loss $\ell(\boldsymbol{\theta}, \boldsymbol{z}) - \ell(\boldsymbol{\theta}_{\text{ref}}, \boldsymbol{z})$, where $\theta$ denotes the target model to be learned, and $\boldsymbol{\theta}_{\text{ref}}$ denotes the reference model, $\boldsymbol{z}$ denotes a data sample, and $\ell(\cdot, \cdot)$ denotes a loss of interest. This approach is intuitive in the sense that a data $\boldsymbol{z}$ with a high loss of the current model $\ell(\boldsymbol{\theta}_t, \boldsymbol{z})$ and a low loss of the reference model $\ell(\boldsymbol{\theta}_{\text{ref}}, \boldsymbol{z})$ means that it has high learnability and should be used for training (Mindermann et al., 2022).

However, the theory behind this approach is quite limited, especially on improving generalization, which hinders our understanding of its effectiveness and derivation of best practice for model steering. Mindermann et al. (2022) tried to motivate this approach through maximizing the likelihood of true labels of classification for a hold-out dataset based on the selection of informative data in the current mini-batch. Through Bayes' theorem and the approximation of negative log-likelihoods by losses, their analysis yields a data selection strategy by selecting data in the mini-batch that have the top values of $\ell(\boldsymbol{\theta}_t, \boldsymbol{z}) - \ell(\boldsymbol{\theta}_{\text{ref}}, \boldsymbol{z})$, which is termed RHO loss selection. While their analysis offers some intuition into why the RHO loss is a reasonable choice for data selection, it falls short of providing guarantees on how and why this approach improves generalization with reduced training data. Moreover, there is no theoretical framework to guide the optimal selection or weighting of data to accelerate training effectively. The heuristic approach that selects data with top RHO losses in the mini-batch is not necessarily the best.

To address this gap, this paper introduces a novel learning framework for enhancing the understanding and practice of model steering based on the RHO loss $\ell(\boldsymbol{\theta}, \boldsymbol{z}) - \ell(\boldsymbol{\theta}_{\text{ref}}, \boldsymbol{z})$. Here, we use the term RHO loss in a broader sense, which is not necessarily restricted to classification as derived in Mindermann et al. (2022). Our framework builds on *Distributionally Robust Optimization (DRO)*. Traditional DRO seeks to minimize the worst-case risk over perturbed data distributions within an uncertainty set derived from the empirical distribution. We extend this idea by applying DRO to the RHO loss $\ell(\boldsymbol{\theta}, \boldsymbol{z}) - \ell(\boldsymbol{\theta}_{\text{ref}}, \boldsymbol{z})$ across the training data, yielding a risk function for model steering, which we term DRRho risk. Leveraging the generalization theory of DRO, we derive theoretical generalization bounds of DRRho risk minimization, offering insights into how it enhances generalization. To the best of our knowledge, this work presents the first generalization theory for model steering, significantly advancing our understanding of its effectiveness.

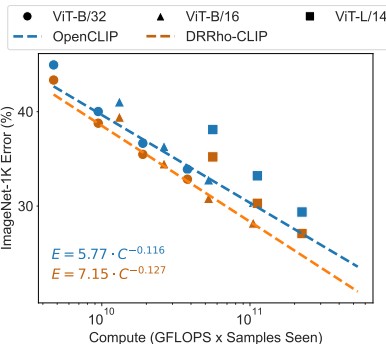

Figure 2: Scaling performance of OpenCLIP (Cherti et al., 2023) and the proposed DRRho-CLIP, which uses the OpenAI CLIP model (Radford et al., 2021) as the reference model. We conduct experiments of the two methods under different settings to fit scaling laws, as shown in the bottom left corner (c.f. Section 6 for more detail).

Our framework not only provides a theoretical foundation for existing heuristic approaches to data selection, sampling, or weighting but also offers improved practices for model steering. To illustrate its practical utility, we focus on contrastive language-image pretraining (CLIP) with a reference model. By leveraging the connection between contrastive loss and DRO (Qiu et al., 2023), we introduce a novel method for CLIP with a reference model, termed DRRho-CLIP, which utilizes the DRRho risk for each anchor data point.

Our experiments demonstrate the effectiveness of DRRho-CLIP. For example, when using OpenAI's CLIP (ViT-B/32) model as a reference, DRRho-CLIP with ViT-B/16 as the backbone trained from scratch on the DFN-192M dataset (Fang et al., 2024) achieves 68.84% zero-shot classification accuracy on ImageNet-1K data. It outperforms OpenAI's CLIP performance of 63.32% trained on a different 400M dataset, and OpenCLIP's performance of 67.8% trained on the same data (cf. Figure 1). In addition, our extensive experiments using various reference models and large-scale datasets reveal the following: (1) DRRho-CLIP achieves comparable performance to the standard CLIP training method (without a reference model) while using significantly less training data (e.g., a 50% reduction). This offers great potential in reducing the burden on collecting high-quality data. (2) DRRho-CLIP outperforms existing heuristic data sampling or selection methods applied to the standard CLIP training framework. (3) When paired with a strong reference model, DRRho-CLIP integrates seamlessly with knowledge distillation, surpassing existing knowledge distillation methods for CLIP training (Vasu et al., 2024). (4) In terms of scaling performance, DRRho-CLIP has a better scaling law than OpenCLIP (Cherti et al., 2023) (cf. Figure 2).

Our **contributions** are summarized as follows:

- We propose a novel framework for model steering based on DRO and th RHO loss. Theoretical generalization bounds are derived and analyzed, which provide insights into why it improves generalization.

- We introduce DRRho-CLIP, a method for training CLIP models with a reference model. Extensive experiments on large-scale datasets and various reference models demonstrate the effectiveness of DRRho-CLIP, including better scaling laws.

## 2. Related Work

Using a pretrained model to improve the training of a target model is not a new idea, which has been long studied in deep learning. A typical approach is transfer learning (Donahue et al., 2013), which uses a pretrained encoder as initialization for finetuning a target model on a different supervised dataset. However, transfer learning restricts the target model to have the same structure (at least in the encoder part) as the pretrained model. There have been some studies trying to expand the architecture of the pretrained model to that of the target model so that the target model can inherit the knowledge of the pretrained model (Chen et al., 2016; Wang et al., 2023; Du et al., 2024). Another category of related works is knowledge distillation, which distills knowledge from a stronger, usually larger, teacher model to a smaller student model (Hinton, 2015). Knowledge distillation has been extensively studied for learning various models (Hinton, 2015; Vasu et al., 2024; Qin et al., 2022). However, these approaches do not consider using the pretrained model for data selection or weighting to facilitate the training of a target model. Hence, the studied technique in this paper is complementary to knowledge distillation.

Leveraging a pretrained model for offline data selection or pruning has been studied. For example, Schuhmann et al. (2021; 2022); Gadre et al. (2023); Fang et al. (2024) leveraged a pretrained model to curate a training dataset to accelerate the training of CLIP models. For each image-text pair in the original dataset, they use the pretrained model (e.g., OpenAI's CLIP model) to compute a score (e.g., similarity score between the image and text) for evaluating its quality. A new subset is then constructed by selecting samples whose score exceeds a certain threshold. Ankner et al. (2024); Marion et al. (2023) have studied a similar idea for pruning the training data for language modeling.

This paper falls into the category of exploiting a pretrained model for online data selection, sampling or weighting. We refer to this learning paradigm as model steering, as the pretrained model is used as a reference to guide training. Mindermann et al. (2022) proposed the RHO loss for selecting samples in a mini-batch for training a classification model. Though similar to offline data curation approaches,

the key differences lie in: (1) a sample that may be discarded in a later stage of training could be useful in the early stage of training; (2) both the model to be trained and the reference model are used together for data selection, sampling, or weighting. The RHO loss has been used in multiple papers (Mindermann et al., 2022; Evans et al., 2025; Lin et al., 2024; Xie et al., 2023; Evans et al., 2024a;b) for training different models. Different from these prior works, our work aims to provide a theoretical foundation for data selection, sampling or weighting with the RHO loss, and use it to derive better practical approaches for training CLIP models. To the best of our knowledge, this is the first work that provides a generalization analysis for learning with a reference model.

Our algorithm for optimizing the proposed contrastive loss informed by DRRho risk is based on recent advances for optimizing global contrastive losses (Yuan et al., 2022; Qiu et al., 2023; Wei et al., 2024). In particular, Yuan et al. (2022) proposed an optimization algorithm termed SogCLR for optimizing a global contrastive loss, which does not suffer from the limitation of using a small mini-batch size. Qiu et al. (2023) has established the connection between DRO and the global contrastive losses, which enables optimization of individual temperature parameters in contrastive learning. Wei et al. (2024) proposed techniques for training CLIP models in a distributed system.

Our theory of DRRho risk minimization is built on existing generalization error bounds for distributionally robust optimization (DRO) (Duchi & Namkoong, 2016; Duchi et al., 2021), which will be reviewed in next section. We notice that a group DRO formulation was considered in (Xie et al., 2023) for using a reference model to learn the weight of a dataset when training language models. However, their formulation does not consider the regularization on the weight variables in DRO, which cannot benefit from the theoretical guarantee as derived in this paper.

## 3. Preliminaries: DRO

**Notations**: Let $z \sim P$ denote a random data, where $P$ represents the distribution of data. We denote by $\boldsymbol{\theta} \in \Theta$ the model parameters, where $\Theta$ is the space of model parameters. Let $\ell(\boldsymbol{\theta}, z)$ denote a loss of interest. The expected risk of $\boldsymbol{\theta}$ is defined as $R(\boldsymbol{\theta}) = \mathbb{E}_{z \sim P}[\ell(\boldsymbol{\theta}, z)]$. Denote by $\boldsymbol{\theta}_* = \arg\min_{\boldsymbol{\theta} \in \Theta} R(\boldsymbol{\theta})$ the optimal solution that minimizes the expected risk. For any model $\boldsymbol{\theta} \in \Theta$, the excess risk $R(\boldsymbol{\theta}) - R(\boldsymbol{\theta}_*)$ is a measure of generalization performance of $\boldsymbol{\theta}$. Let $\mathcal{F} = \{\ell(\boldsymbol{\theta}, \cdot), \boldsymbol{\theta} \in \Theta\}$. To derive the generalization error bounds, we need some complexity measure of the function class $\mathcal{F}$. We follow Duchi & Namkoong (2016) by using the VC-dimension of $\mathcal{F}$ denoted by $d_v = \mathrm{VC}(\mathcal{F})$, which is defined as VC-dimension of the set of subgraphs of functions in $\mathcal{F}$ (van der Vaart & Wellner, 1996).

Given $n$ samples $\boldsymbol{z}_1, \ldots, \boldsymbol{z}_n$ drawn i.i.d. from $P$, DRO solves the following problem:

$$\hat{\boldsymbol{\theta}}_* \in \arg\min_{\boldsymbol{\theta} \in \Theta} \sup_{\substack{\mathbf{p} \in \Delta \\ D_\phi(\mathbf{p}, 1/n) \le \rho/n}} \sum_{i=1}^n p_i \ell(\boldsymbol{\theta}, \boldsymbol{z}_i), \quad (1)$$

where $\Delta = \{\mathbf{p} \in \mathbb{R}^n : \mathbf{p} \ge 0, \sum_{i=1}^n p_i = 1\}$ is a simplex, $D_\phi(\mathbf{p}, 1/n) = \frac{1}{n}\sum_{i=1}^n \phi(p_i n)$ denotes the $\phi$-divergence between $\mathbf{p}$ and uniform probabilities, (e.g., Kullback-Leibler divergence (Kullback, 1997), CVaR-divergence (Levy et al., 2020), $\chi^2$-divergence (Duchi & Namkoong, 2016)), and $\rho \ge 0$ is a hyperparameter. DRO minimizes a worst-case risk over all possible perturbed distributions from the empirical distribution within an uncertainty set. The supremum over the weights $\mathbf{p}$ indicates that if a sample is hard (i.e., has a large loss value), then it will have a large weight so that the algorithm will pay more attention to optimizing it.

The generalization error bounds of DRO have been developed (Duchi & Namkoong, 2016; Duchi et al., 2021). We present a corollary from Duchi & Namkoong, Theorem 3.

**Theorem 3.1.** *Let $D_\phi$ be $\chi^2$-divergence with $\phi(t) = (t-1)^2/2$. Assume that $\ell(\boldsymbol{\theta}, \cdot) \in [M_0, M_1]$ with $M = M_1 - M_0$. Let $C_1 = \log\frac{1}{\delta} + \log c + d_v \log 16e + 2d_v \log n$ for a given $\delta > 0$ and some constant $c < \infty$. If $C_1 \cdot n \ge 8M^2$ and $\rho \ge 9C_1$, then with probability at least $1 - \delta$,*

$$R(\hat{\boldsymbol{\theta}}_*) \le \inf_{\boldsymbol{\theta} \in \Theta} \left( R(\boldsymbol{\theta}) + 2\sqrt{\frac{2\rho}{n}\mathrm{Var}(\ell(\boldsymbol{\theta}, \cdot))} \right) + \frac{C_2}{n},$$

*where $\mathrm{Var}(\ell(\boldsymbol{\theta}, \cdot))$ denotes the variance of $\ell(\boldsymbol{\theta}, \boldsymbol{z})$ for $\boldsymbol{z} \sim P$ and $C_2 = (25\rho/3 + 2)M$.*

The above theorem indicates that DRO may have a better excess risk $R(\hat{\boldsymbol{\theta}}_*) - R(\boldsymbol{\theta}_*)$ than that of ERM when the variance term $\mathrm{Var}(\ell(\boldsymbol{\theta}_*, \cdot))$ at the optimal solution $\boldsymbol{\theta}_*$ is small (Duchi & Namkoong, 2016). Although the above result is derived for the $\chi^2$-divergence, one can follow Duchi et al. (2021) to derive similar results for other divergence, e.g., KL-divergence. Nevertheless, we focus on the theoretical insights that DRO can bring to our framework.

## 4. DRRho Risk Minimization

A central argument for the improved generalization bound of DRO is when the variance $\mathrm{Var}(\ell(\boldsymbol{\theta}_*, \cdot))$ is small. However, achieving a low variance is not always guaranteed. To address this, we propose to leverage a reference model $\boldsymbol{\theta}_{\mathrm{ref}}$ to reduce this variance. Specifically, our framework of model steering involves replacing the standard loss function $\ell(\boldsymbol{\theta}, \cdot)$ with the RHO loss, defined as $\hat{\ell}(\boldsymbol{\theta}, \cdot) = \ell(\boldsymbol{\theta}, \cdot) - \ell(\boldsymbol{\theta}_{\mathrm{ref}}, \cdot)$. In particular, we define the following risk:

$$F(\boldsymbol{\theta}) := \sup_{\substack{\mathbf{p} \in \Delta \\ D_\phi(\mathbf{p}, 1/n) \le \rho/n}} \sum_{i=1}^n p_i(\ell(\boldsymbol{\theta}, \boldsymbol{z}_i) - \ell(\boldsymbol{\theta}_{\mathrm{ref}}, \boldsymbol{z}_i)). \quad (2)$$

We refer to $F(\boldsymbol{\theta})$ as the DRRho risk. Building upon this, we formulate the DRRho risk minimization problem:

$$\tilde{\boldsymbol{\theta}}_* \in \arg\min_{\boldsymbol{\theta} \in \Theta} F(\boldsymbol{\theta}). \quad (3)$$

Next, we present the generalization error bounds of DRRho risk minimization. We consider the function class $\mathcal{F}_r = \{\ell(\boldsymbol{\theta}, \cdot) - \ell(\boldsymbol{\theta}_{\mathrm{ref}}, \cdot), \boldsymbol{\theta} \in \Theta\}$ and abuse the notation $d_v = \mathrm{VC}(\mathcal{F}_r)$. The proofs of the following results are presented in Appendix A.

**Theorem 4.1.** *Under the same setting of Theorem 3.1, let $C_1 = \log\frac{1}{\delta} + \log c + d_v \log 16e + 2d_v \log n$ for a given $\delta > 0$ and some constant $c < \infty$. If $C_1 \cdot n \ge 32M^2$ and $\rho \ge 9C_1$, with probability at least $1 - \delta$,*

$$R(\tilde{\boldsymbol{\theta}}_*) \le \inf_{\boldsymbol{\theta} \in \Theta} \left( R(\boldsymbol{\theta}) + \sqrt{\frac{2\rho}{n}\mathrm{Var}(\ell(\boldsymbol{\theta}, \cdot) - \ell(\boldsymbol{\theta}_{\mathrm{ref}}, \cdot))} \right) + \frac{C_2}{n},$$

*where $C_2 = (50\rho/3 + 4)M$.*

We present two corollaries to understand how DRRho risk minimization improves the generalization over DRO / ERM.

**Corollary 4.2.** *Under the same setting of Theorem 4.1, we have*

$$R(\tilde{\boldsymbol{\theta}}_*) \le R(\boldsymbol{\theta}_*) + \sqrt{\frac{2\rho}{n}\mathrm{Var}(\ell(\boldsymbol{\theta}_*, \cdot) - \ell(\boldsymbol{\theta}_{\mathrm{ref}}, \cdot))} + \frac{C_2}{n}.$$

**Remark:** Comparing the excess risk bound of $R(\tilde{\boldsymbol{\theta}}_*) - R(\boldsymbol{\theta}_*)$ for DRRho with that of $R(\hat{\boldsymbol{\theta}}_*) - R(\boldsymbol{\theta}_*)$ for DRO, we can see that the variance term changes to $\mathrm{Var}(\ell(\boldsymbol{\theta}_*, \cdot) - \ell(\boldsymbol{\theta}_{\mathrm{ref}}, \cdot))$ from $\mathrm{Var}(\ell(\boldsymbol{\theta}_*, \cdot))$. It is reasonable to assume that the reference model $\boldsymbol{\theta}_{\mathrm{ref}}$ is sufficiently trained such that $\ell(\boldsymbol{\theta}_{\mathrm{ref}}, \cdot)$ has a similar distribution to $\ell(\boldsymbol{\theta}_*, \cdot)$; hence we expect that $\mathrm{Var}(\ell(\boldsymbol{\theta}_*, \cdot) - \ell(\boldsymbol{\theta}_{\mathrm{ref}}, \cdot))$ would be much smaller than $\mathrm{Var}(\ell(\boldsymbol{\theta}_*, \cdot))$.

The following corollary provides insights into the reduced sample complexity of DRRho risk minimization in achieving the same level of generalization as the reference model.

**Corollary 4.3.** *Under the same setting of Theorem 4.1, and assume $\boldsymbol{\theta}_{\mathrm{ref}} \in \Theta$, we have*

$$R(\tilde{\boldsymbol{\theta}}_*) - R(\boldsymbol{\theta}_*) \le R(\boldsymbol{\theta}_{\mathrm{ref}}) - R(\boldsymbol{\theta}_*) + \frac{C_2}{n}.$$

**Remark:** The above result allows us to compare the excess risk of DRRho risk minimizer $\tilde{\boldsymbol{\theta}}_*$ with that of a reference model $\boldsymbol{\theta}_{\mathrm{ref}} \in \Theta$ that is from the same family. Suppose that reference model $\boldsymbol{\theta}_{\mathrm{ref}}$ is learned using ERM on a dataset of $m$ samples. A standard generalization error analysis (Boucheron et al., 2005) would yield an excess risk bound on the level of $\mathcal{O}(\sqrt{1/m})$, i.e., $R(\boldsymbol{\theta}_{\mathrm{ref}}) - R(\boldsymbol{\theta}_*) = \mathcal{O}(1/\sqrt{m})$. In order to reach the same level of generalization error of the reference model, DRRho needs only

$n = \mathcal{O}(\sqrt{m})$ samples, which dramatically reduces the sample complexity $\mathcal{O}(m)$ of ERM without a reference model.

Before ending this section, we discuss how DRRho risk minimization provides a foundation for existing heuristic approaches for data selection, weighting, and sampling based on the RHO loss, and for inducing better practices.

**Data Selection:** When we use the CVaR divergence $\phi(t) = 1$ if $t \leq n/k$, and $\phi(t) = \infty$ otherwise, the DRRho risk becomes the average of top-$k$ RHO losses:

$$F(\boldsymbol{\theta}) := \frac{1}{k} \sum_{i=1}^{k} \ell(\boldsymbol{\theta}, \boldsymbol{z}_{[i]}) - \ell(\boldsymbol{\theta}_{\text{ref}}, \boldsymbol{z}_{[i]}). \qquad (4)$$

where $\boldsymbol{z}_{[i]}$ denotes the data whose RHO loss is ranked at the $i$-th position in descending order. Existing studies have applied RHO-loss-based data selection to the mini-batch for simplicity (Mindermann et al., 2022; Lin et al., 2024), which do not necessarily optimize the average of top-$k$ RHO loss among the whole dataset as in our framework.

**Data Weighting / Sampling:** If we use the KL-divergence $\text{KL}(\mathbf{p}, \mathbf{1}/n) = \sum_{i=1}^{n} p_i \log(p_i n)$, where $\phi(t) = t \log t - t + 1$, then from the Lagrange dual theory we can derive:

$$F(\boldsymbol{\theta}) := \min_{\tau \geq 0} \tau \log \left( \frac{1}{n} \sum_{i=1}^{n} \exp(\frac{(\ell(\boldsymbol{\theta}, \boldsymbol{z}_i) - \ell(\boldsymbol{\theta}_{\text{ref}}, \boldsymbol{z}_i)}{\tau}) \right)$$
$$+ \tau \rho / n. \qquad (5)$$

If we compute the gradient of the above objective in terms of $\boldsymbol{\theta}$ given $\tau$, we obtain $\sum_{i=1}^{n} p_i \nabla_{\boldsymbol{\theta}} \ell(\boldsymbol{\theta}, \boldsymbol{z}_i)$, where

$$p_i = \frac{\exp\left(\frac{\ell(\boldsymbol{\theta}, \boldsymbol{z}_i) - \ell(\boldsymbol{\theta}_{\text{ref}}, \boldsymbol{z}_i)}{\tau}\right)}{\sum_{j=1}^{n} \exp\left(\frac{\ell(\boldsymbol{\theta}, \boldsymbol{z}_j) - \ell(\boldsymbol{\theta}_{\text{ref}}, \boldsymbol{z}_j)}{\tau}\right)}.$$

Hence, the above DRRho risk acts like assigning different data different weights such that data with a larger RHO loss has a higher weight in the gradient calculation. Heuristic approaches have implemented this idea by sampling data in a large batch following $\mathbf{p}_i$ calculated based on the mini-batch data to create a smaller batch (Evans et al., 2024a;b).

To simplify the complexity of optimization, one can turn $\tau$ into a hyperparameter, which is equivalent to using a KL divergence as regularization in defining the DRRho risk:

$$F(\boldsymbol{\theta}) = \sup_{\mathbf{p} \in \Delta} \sum_{i=1}^{n} p_i (\ell(\boldsymbol{\theta}, \boldsymbol{z}_i) - \ell(\boldsymbol{\theta}_{\text{ref}}, \boldsymbol{z}_i)) - \tau \text{KL}(\mathbf{p}, \mathbf{1}/n)$$
$$= \tau \log \left( \frac{1}{n} \sum_{i=1}^{n} \exp(\frac{(\ell(\boldsymbol{\theta}, \boldsymbol{z}_i) - \ell(\boldsymbol{\theta}_{\text{ref}}, \boldsymbol{z}_i)}{\tau}) \right).$$
$$(6)$$

Finally, we would like to mention that efficient stochastic algorithms have been developed to optimize the objectives in Equations (4) to (6) (Qi et al., 2023b;a; Wang & Yang, 2023; Hu et al., 2023). Hence, solving the proposed DRRho risk minimization with these advanced optimization algorithms could yield better guarantees than heuristic approaches in existing studies. We illustrate this with an application of CLIP training in next section.

## 5. DRRho-CLIP with a Reference Model

In this section, we explore the application of the DRRho risk to CLIP training (Radford et al., 2021) with a reference model. Although the proposed DRRho risk minimization framework is general and can be applied to training various models, we focus on CLIP training for several reasons: (i) CLIP involves more complex data structures, including anchor data and their corresponding negative samples for contrastive learning. Without guidance from theory, heuristic methods that sample data based on the RHO loss may not give optimal performance (Evans et al., 2024a). (ii) The publicly available OpenAI's CLIP model has already been utilized to enhance CLIP training by filtering high-quality data based on the CLIP score (Gadre et al., 2023). This raises an intriguing question: can our framework deliver additional improvements when using OpenAI's CLIP model as a reference on the filtered data?

For CLIP, the training dataset consists of images and their corresponding text descriptions. We use $\boldsymbol{x}$ to denote an image and $\boldsymbol{y}$ to denote a text, and use $\boldsymbol{z} = (\boldsymbol{x}, \boldsymbol{y})$ to denote a pair. We use $\mathcal{S} = \{(\boldsymbol{x}_1, \boldsymbol{y}_1), \ldots, (\boldsymbol{x}_n, \boldsymbol{y}_n)\}$ to denote a training set of size $n$. Given an image $\boldsymbol{x}_i$, let $\boldsymbol{e}_{1,i} = \boldsymbol{e}_1(\boldsymbol{\theta}_1, \boldsymbol{x}_i) \in \mathbb{R}^d$ denote image representation by an image encoder with parameter $\boldsymbol{\theta}_1$. Similarly, $\boldsymbol{e}_{2,i} = \boldsymbol{e}_2(\boldsymbol{\theta}_2, \boldsymbol{y}_i) \in \mathbb{R}^d$ denotes the embedding of text $\boldsymbol{y}_i$ by the text encoder with parameter $\boldsymbol{\theta}_2$. Let $\boldsymbol{\theta} = (\boldsymbol{\theta}_1, \boldsymbol{\theta}_2)$ denote the parameters of the image encoder and the text encoder jointly. Let $\mathcal{S}_{i-} = \mathcal{S} \backslash \{i\}$ denote the dataset without $i$-th pair. Let $s(\boldsymbol{x}_i, \boldsymbol{y}_j)$ denote the cosine similarity between $i$-th image embedding $\boldsymbol{e}_{1,i}$ and $j$-th text embedding $\boldsymbol{e}_{2,j}$.

In order to apply DRRho risk to CLIP, we leverage the connection between a contrastive loss and DRO (Qiu et al., 2023). For simplicity of our presentation, we consider the KL-regularized DRO formulation (6), which yields an objective with a tunable temperature parameter $\tau$. Similar extensions can be made to KL-constrained DRO (5) for learnable temperature (cf. Appendix C.3). First we define the contrastive loss function for each anchor image data $\boldsymbol{x}_i$ without using a reference model. To this end, we define a pairwise loss $\ell(\boldsymbol{\theta}, \boldsymbol{x}_i, \boldsymbol{y}_j) = s(\boldsymbol{x}_i, \boldsymbol{y}_j) - s(\boldsymbol{x}_i, \boldsymbol{y}_i)$, which measures the gap of similarities between a negative pair $(\boldsymbol{x}_i, \boldsymbol{y}_j)$ and the positive pair $(\boldsymbol{x}_i, \boldsymbol{y}_i)$ of the anchor data $\boldsymbol{x}_i$. Then we use DRO to aggregate these individual pairwise

losses for $\boldsymbol{y}_j \in \mathcal{S}$ into the following loss of $\boldsymbol{x}_i$:

$$F_{\mathrm{dro}}(\boldsymbol{\theta}, \boldsymbol{x}_i, \mathcal{S}) := \max_{\boldsymbol{p} \in \Delta} \sum_{j=1}^{n} p_j \ell(\boldsymbol{\theta}, \boldsymbol{x}_i, \boldsymbol{y}_j) - \tau \mathrm{KL}(\boldsymbol{p}, \mathbf{1}/n),$$

$$= \tau \log \left( \frac{1}{n} \sum_{j=1}^{n} \exp(\frac{\ell(\boldsymbol{\theta}, \boldsymbol{x}_i, \boldsymbol{y}_j)}{\tau}) \right). \quad (7)$$

To apply DRRho risk for integrating a reference model, we plug the following shifted loss into the above formulation:

$$\hat{\ell}(\boldsymbol{\theta}, \boldsymbol{\theta}_{\mathrm{ref}}, \boldsymbol{x}_i, \boldsymbol{y}_j) := \ell(\boldsymbol{\theta}, \boldsymbol{x}_i, \boldsymbol{y}_j) - \ell(\boldsymbol{\theta}_{\mathrm{ref}}, \boldsymbol{x}_i, \boldsymbol{y}_j).$$

As a result, we obtain the following DRRho contrastive loss for each image $\boldsymbol{x}_i$:

$$F(\boldsymbol{\theta}, \boldsymbol{x}_i, \mathcal{S}) = \tau \log \left( \frac{1}{n} \sum_{j=1}^{n} \exp(\frac{\hat{\ell}(\boldsymbol{\theta}, \boldsymbol{\theta}_{\mathrm{ref}}, \boldsymbol{x}_i, \boldsymbol{y}_j)}{\tau}) \right). \quad (8)$$

Similarly, we define a DRRho contrastive loss for text $\boldsymbol{y}_i$:

$$F(\boldsymbol{\theta}, \boldsymbol{y}_i, \mathcal{S}) = \tau \log \left( \frac{1}{n} \sum_{j=1}^{n} \exp(\frac{\hat{\ell}(\boldsymbol{\theta}, \boldsymbol{\theta}_{\mathrm{ref}}, \boldsymbol{y}_i, \boldsymbol{x}_j)}{\tau}) \right) \quad (9)$$

Then, we solve the following optimization problem:

$$\min_{\boldsymbol{\theta}} \frac{1}{n} \sum_{i=1}^{n} (F(\boldsymbol{\theta}, \boldsymbol{x}_i, \mathcal{S}) + F(\boldsymbol{\theta}, \boldsymbol{y}_i, \mathcal{S})).$$

To optimize the above objective, we use the SogCLR algorithm (Yuan et al., 2022) that has a provable convergence guarantee without using a large batch size. In particular, the algorithm maintains two sequences of estimators $u_{1,i}^t, u_{2,i}^t, t = 1, \ldots, T$ to track the inner average in the log function of $F(\boldsymbol{\theta}, \boldsymbol{x}_i, \mathcal{S})$ and $F(\boldsymbol{\theta}, \boldsymbol{y}_i, \mathcal{S})$. At $t$-th iteration, the following update with mini-batch $\mathcal{B}^t$ are executed:

$$u_{1,i}^{t+1} = (1 - \gamma_t)u_{1,i}^t + \gamma_t \mathbb{E}_{j \in \mathcal{B}_{i-}^t} \exp(\frac{\hat{\ell}_1(\boldsymbol{\theta}_t, \boldsymbol{\theta}_{\mathrm{ref}}, \boldsymbol{x}_i, \boldsymbol{y}_j)}{\tau}),$$

$$u_{2,i}^{t+1} = (1 - \gamma_t)u_{2,i}^t + \gamma_t \mathbb{E}_{j \in \mathcal{B}_{i-}^t} \exp(\frac{\hat{\ell}_2(\boldsymbol{\theta}_t, \boldsymbol{\theta}_{\mathrm{ref}}, \boldsymbol{y}_i, \boldsymbol{x}_j)}{\tau}), \quad (10)$$

where $\mathcal{B}_{i-}^t = \mathcal{B}^t \setminus \{i\}$ and $\mathbb{E}_{j \in \mathcal{B}_{i-}^t}$ denotes the average over data in $\mathcal{B}_{i-}^t$, and $\gamma_t$ is regarded as an inner learning rate. Then we compute the gradient estimators by

$$G_1^t = \mathbb{E}_{i \in \mathcal{B}^t} \frac{\tau}{\varepsilon + u_{1,i}^{t+1}} \cdot \nabla_{\boldsymbol{\theta}} \mathbb{E}_{j \in \mathcal{B}_{i-}^t} \exp(\frac{\hat{\ell}_1(\boldsymbol{\theta}_t, \boldsymbol{\theta}_{\mathrm{ref}}, \boldsymbol{x}_i, \boldsymbol{y}_j)}{\tau}),$$

$$G_2^t = \mathbb{E}_{i \in \mathcal{B}^t} \frac{\tau}{\varepsilon + u_{2,i}^{t+1}} \cdot \nabla_{\boldsymbol{\theta}} \mathbb{E}_{j \in \mathcal{B}_{i-}^t} \exp(\frac{\hat{\ell}_2(\boldsymbol{\theta}_t, \boldsymbol{\theta}_{\mathrm{ref}}, \boldsymbol{y}_i, \boldsymbol{x}_j)}{\tau}). \quad (11)$$

where $\varepsilon$ is treated as a hyperparameter in practice (Wei et al., 2024). We present the details of our algorithm in Algorithm 1, which is referred to as DRRho-CLIP.

---

**Algorithm 1** DRRho-CLIP

1: **Input:** Model $\boldsymbol{\theta}^0, \tau^0$, sequences $\{u_{1,i}^0\}, \{u_{2,i}^0\}$, number of iterations $T$.
2: **for** $t = 0, \ldots, T - 1$ **do**
3:     Sample a batch $\mathcal{B}^t \subset \mathcal{S}$ and compute features
4:     Update $u_{1,i}^{t+1}, u_{2,i}^{t+1}$ using (10) for all $i \in \mathcal{B}^t$
5:     Set $u_{1,i}^{t+1} = u_{1,i}^t, u_{2,i}^{t+1} = u_{2,i}^t$ for $i \notin \mathcal{B}^t$
6:     Compute $G_1^t, G_2^t$ using Equation (11)
7:     Update $\boldsymbol{\theta}_{t+1}$ using $G_1^t + G_2^t$ as a gradient estimator with an optimizer (e.g., AdamW)
8: **end for**

---

**Efficient Implementation:** It is notable that, like other LEAR methods, DRRho-CLIP requires computing the loss of the reference model on the training data. It is expensive to compute these losses on the fly. To reduce this cost, we compute the embedding vectors of training data by the reference model in an offline manner. During training, we load the pre-computed features of the current mini-batch from the disk and compute the loss based on the pre-computed features. Similar approaches have been used in (Vasu et al., 2024; Evans et al., 2024a).

## 6. Experiments

In this section, we conduct experiments to demonstrate the superiority of our proposed framework, where we focus on training CLIP models. First, we empirically verify claims of our theory: (1) our framework is more data-efficient than learning without a reference model, and (2) the variance of the RHO loss in the excess risk bound of our framework is lower than that of a regular loss. Next, we compare DRRho-CLIP with other baselines, and we also show that DRRho-CLIP can be seamlessly integrated with distillation methods. Finally, we study the scaling law of DRRho-CLIP and show that it has a better scaling trend than OpenCLIP.

- **Data**: The training datasets we use consist of CC12M (9M samples) (Changpinyo et al., 2021), DFN-192M (192M samples) (Fang et al., 2024) and DFN-12M (a 12M subset selected from DFN-192M).

- **Models**: The reference model is either ViT-B/32, ViT-B/16 or ViT-L/14, which are either pretrained open-weight models or models we trained from scratch. For ease of presentation, we use "model (data)" to denote a model and the data on which it is pretrained. The following reference models are pretrained open-weight models: ViT-B/32 (WIT-400M) (Radford et al., 2021), ViT-B/16 (DFN-2B) and ViT-L/14 (DFN-2B) (Fang et al., 2024). The target model trained from scratch could be ViT-B/32 or ViT-B/16.

- **Metrics**: We leverage the Datacomp benchmark (Gadre et al., 2023) to evaluate the performance of target models, which comprises 38 zero-shot classification and retrieval

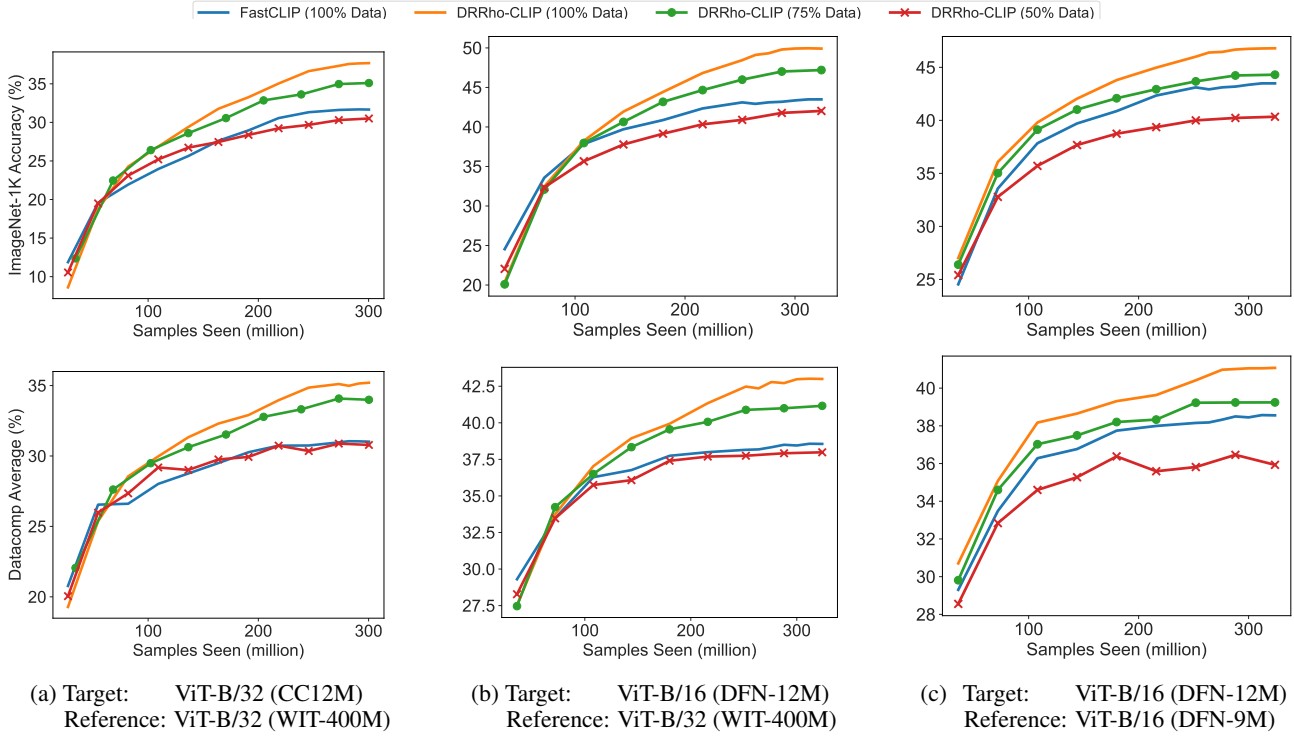

(a) Target:     ViT-B/32 (CC12M)     (b) Target:     ViT-B/16 (DFN-12M)     (c) Target:     ViT-B/16 (DFN-12M)
Reference: ViT-B/32 (WIT-400M)     Reference: ViT-B/32 (WIT-400M)     Reference: ViT-B/16 (DFN-9M)

Figure 3: Performance curves of FastCLIP and DRRho-CLIP with different target and reference models (with **each column** representing one combination). **Top row**: ImageNet Top-1 accuracy, **bottom row**: Datacomp average performance.

tasks. The numbers we report are the ImageNet-1K Top 1 accuracy and the average performance on the 38 tasks.

- **Hardware and Training framework**: Unless otherwise specified, we use a batch size of 5120 on 8 H100 GPUs for training ViT-B/16, and use a batch size of 4096 on 8 A100 GPUs for training ViT-B/32. We implement our method using the codebase of FastCLIP distributed training framework (Wei et al., 2024). For hyperparameter tuning, we refer readers to Appendix C.2.

### 6.1. Empirical Verification of Theoretical Results

**DRRho-CLIP is more data-efficient**. Corollary 4.3 implies that DRRho requires less amount of training data than learning without a reference model to achieve the same level of excess risk. To empirically verify this theory, we run DRRho-CLIP on different portions of the training dataset (100%, 75% and 50%) and compare it with the baseline FastCLIP (Wei et al., 2024) on the whole training dataset (100%). Multiple target / reference model combination are tested, as specified in the following table.

| Target Model (Data) | Reference Model (Data) |
| --- | --- |
| ViT-B/32 (CC12M) | ViT-B/32 (WIT-400M) |
| ViT-B/16 (DFN-12M) | ViT-B/32 (WIT-400M) |
| ViT-B/16 (DFN-12M) | ViT-B/16 (DFN-9M) |

The performance curves of different methods on different metrics are plotted in Figure 3. From the results we can observe that (i) with OpenAI's ViT-B/32 (WIT-400M) as the reference model, DRRho-CLIP with 50% of training data can achieve comparable performance to FastCLIP on the whole training data (Figure 3a, b); (ii) with 100% training data, DRRho-CLIP outperforms FastCLIP by a large margin; (iii) even with a weaker reference model ViT-B/16 (DFN-9M) (Figure 3c) trained by us using FastCLIP on a 9M subset of DFN-12M, DRRho-CLIP still benefits from it, achieving better performance with 75% data (9M) than FastCLIP trained on DFN-12M without a reference model.

**DRRho-CLIP has a lower variance of RHO loss**. From Corollary 4.2 and the remark below we know that the main difference between the excess risk bound of DRO and that of DRRho lies in the variance term $(\mathrm{Var}(\ell(\boldsymbol{\theta}_*,\cdot))$ for DRO and $\mathrm{Var}(\ell(\boldsymbol{\theta}_*,\cdot) - \ell(\boldsymbol{\theta}_{\mathrm{ref}},\cdot))$ for DRRho). To have a better understanding of the two terms, we train a ViT-B/16 target model on DFN-12M with 320M samples seen using FastCLIP and DRRho-CLIP with a reference model ViT-B/32 (WIT-400M). Then we select a 200K subset from the training data and compute the variance of the original pairwise loss in FastCLIP and the RHO loss in DRRho-CLIP w.r.t. each image and text. We use the trained model for computing the variance of the RHO loss and the original.

Table 1: Comparison of different methods on DFN-192M with 1.28B samples seen. Reference denotes the performance of the reference model. OpenCLIP and FastCLIP does not leverage a reference model. For distillation-based methods, the reference model is also the teacher model for distillation.

| Metric | Method | Reference Model | | |
| --- | --- | --- | --- | --- |
| | | ViT-B/32 (WIT-400M) | ViT-B/16 (DFN-2B) | ViT-L/14 (DFN-2B) |
| ImageNet Top 1 | Reference | 63.32 | 76.23 | 81.41 |
| | OpenCLIP | 66.94 | 66.94 | 66.94 |
| | FastCLIP | 67.37 | 67.37 | 67.37 |
| | JEST | 56.40 | 56.56 | 55.96 |
| | JEST (Top-k) | 57.75 | 57.34 | 56.96 |
| | DRRho-CLIP | **68.84** | **69.19** | **68.69** |
| | MobileCLIP (w/ Distillation) | 66.94 | 68.67 | 68.47 |
| | FastCLIP (w/ Distillation) | 67.33 | 69.15 | 68.85 |
| | DRRho-CLIP (w/ Distillation) | **68.84** | **69.50** | **69.25** |
| Datacomp | Reference | 52.27 | 60.75 | 66.65 |
| | OpenCLIP | 54.58 | 54.58 | 54.58 |
| | FastCLIP | 54.69 | 54.69 | 54.69 |
| | JEST | 48.25 | 48.97 | 48.27 |
| | JEST (Top-k) | 48.26 | 49.22 | 48.67 |
| | DRRho-CLIP | **55.20** | **55.48** | **54.74** |
| | MobileCLIP (w/ Distillation) | 54.58 | 55.21 | 55.31 |
| | FastCLIP (w/ Distillation) | 54.69 | 55.60 | 55.91 |
| | DRRho-CLIP (w/ Distillation) | **55.20** | **57.17** | **56.29** |

| Ref. Model | Loss Variance ($\times 10^{-3}$) | | DC |
| --- | --- | --- | --- |
| | Image | Text | |
| No Ref. | 7.26 (0.58) | 7.02 (0.91) | 38.57 |
| ViT-B/32 (WIT-400M) | 4.49 (0.54) | 4.09 (0.60) | 43.02 |

We report the mean and standard deviation of the variance above, along with the Datacomp (DC) average performance of both methods. The results show that with a reference model, the variance of the RHO loss is lower than that of the original loss.

### 6.2. Comparison with Baselines

In this section, we compare our method with other baselines of learning with reference models and knowledge distillation. For the former, we compare with JEST (Evans et al., 2024a), which is a heuristic approach of applying the RHO loss for data sampling. In particular, they first sample image-text pairs from a large batch according to their similarities, and then compute an averaged RHO loss for each remaining data using the selected data in previous step as negative data and then perform sampling. We also implement another variant that chooses the top pairs in the remaining data based on their averaged RHO loss in the second step, which is referred to as JEST (Top-$k$). Then a mini-batch contrastive loss is computed based on the selected data for updating the model parameter. These methods are implemented to select the same size of mini-batch samples as our method from a 5

times larger super-batch.

For knowledge distillation, we compare with Mobile-CLIP (Vasu et al., 2024), which optimizes:

$$\min_{\boldsymbol{\theta} \in \Theta} (1 - \lambda)\mathcal{L}_{\text{con}}(\boldsymbol{\theta}) + \lambda\mathbb{E}_{\mathcal{B} \subset \mathcal{S}}\mathcal{L}_{\text{dist}}(\boldsymbol{\theta}, \boldsymbol{\theta}_{\text{ref}}, \mathcal{B}). \quad (12)$$

where $\lambda \in [0, 1]$ is a hyperparameter, $\mathcal{L}_{\text{con}}$ denotes a mini-batch contrastive loss and $\mathcal{L}_{\text{dist}}$ is the distillation loss between the target model and reference model (cf. Equation (16) in Appendix C). To demonstrate the benefit of our approach integrated with knowledge distillation, we replace the contrastive loss above with our DRRho contrastive loss, which is referred to as DRRho-CLIP (w/Distillation). For comparison, we also implement another baseline Fast-CLIP (w/Distillation) which uses the global contrastive loss instead of the mini-batch contrastive loss as in MobileCLIP.

We present the results on DFN-192M in Table 1 (results on DFN-12M are presented in Table 3 in Appendix C due to space limit). All methods reported train a target model of ViT-B/16 from scratch. From the results we arrive at the following conclusions: (1) Compared with JEST and JEST (Top-k), our approach achieves much better performance. This shows that our theory-driven approach is better than the heuristic approaches. (2) When the reference model is relatively weak, e.g., ViT-B/32 (WIT-400M), DRRho-CLIP achieves better performance (68.84% ImageNet Top-1 accuracy) than MobileCLIP (66.94%) and the reference model itself (cf. Figure 4); (3) when the reference models

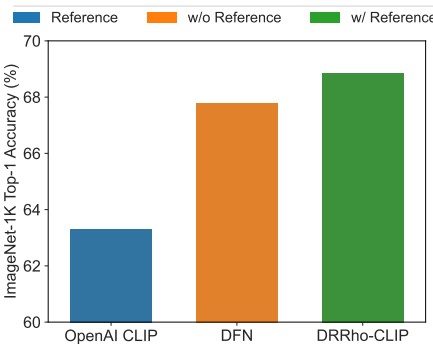

Figure 4: Zero-shot Top 1 Accuracy on ImageNet-1K of different models. DFN model was trained on DFN-192M dataset with 1.28B samples seen with batch size 8192 (Fang et al., 2024), DRRho-CLIP model was trained under the same setting with batch size 5120, and using OpenAI CLIP as a reference model.

are strong, e.g., ViT-B/16 (DFN-2B), ViT-L/14 (DFN-2B), DRRho-CLIP (w/ Distillation) achieves the best result.

### 6.3. Scaling Law

Finally, we study the scaling law of DRRho-CLIP, running with a batch size of 5120 on 8 H100 GPUs and using ViT-B/32 (WIT-400M) as the reference model. Similar to Cherti et al. (2023), we aim at uncovering the scaling law in the following form: $E = \alpha \cdot C^\beta$, where $E$ denotes the ImageNet error rate, $C$ denotes the amount of compute (GFLOPs), $\alpha, \beta$ are real numbers that need to be determined. We choose number of samples seen $T = 0.32, 0.64, 1.28, 2.56$B for our experiments, and use ViT-B/32, ViT-B/16 and ViT-L/14 as the target model. We follow the open_clip repository and use its calculated amount of compute for each model[2].

To estimate $E$ for each $C$, we run our method with ViT-B/16 on datasets of varying sizes from 144M to 624M, which are subsets selected from DFN-2B (Fang et al., 2024), and use the lowest error rate among different dataset sizes to estimate $E$. Next, we run our method with ViT-B/32 and ViT-L/14 on datasets of the same sizes as in ViT-B/16 training. Then we fit the power law with different $C$ and corresponding $E$. We repeat the same procedure for OpenCLIP. We plot the relationship $\log E = \log \alpha + \beta \log C$ in Figure 2 for both DRRho-CLIP and OpenCLIP. We observe that DRRho-CLIP has a better scaling law than OpenCLIP with smaller $\beta$. This also demonstrates that our theory has practical implications. We would like to note that in Figure 2 we do not account for the compute of the reference model (mostly for obtaining image and text features), since these features are usually already computed and stored for other purposes and can thus be leveraged free of charge (e.g., Common-Pool (Gadre et al., 2023) and LAION-5B (Schuhmann et al.,

___
[2]github.com/mlfoundations/open_clip/docs/model_profile.csv

2022) use these features to filter out low-quality data). Nevertheless, we present the results of taking the compute of the reference model into consideration in Appendix C.5, where we observe similar trend.

## 7. Conclusion

In this paper, we have presented a novel learning paradigm of model steering, which uses a reference model to guide the training. Different from other heuristic approaches for learning with a reference model, our framework is grounded on distributionally robust optimization theory. We provided generalization analysis for our framework to analyze the improved generalization and data efficiency of our method. We applied our approach to CLIP training and proposed DRRho-CLIP. Experiments of DRRho-CLIP not only verified the theory but also demonstrated its superiority over heuristic approaches and a better scaling law than that of the state-of-the-art CLIP training method.

## Acknowledgments

We thank anonymous reviewers for constructive comments. This work used GPU resources at TAMU ACES and NCSA Delta through allocation CIS230245 from the Advanced Cyberinfrastructure Coordination Ecosystem: Services & Support (ACCESS) program, which is supported by U.S. National Science Foundation grants #2138259, #2138286, #2138307, #2137603, and #2138296. XW and TY were partially supported by National Science Foundation Award #2306572 and #2147253, National Institutes of Health Award #R01HL168116. FY and FS were partially supported by National Science Foundation Award #2326495 and #2247060, and Lilly Endowment, Inc., through its support for the Indiana University Pervasive Technology Institute.

## Impact Statement

This paper presents work whose goal is to advance the field of Machine Learning. There are many potential societal consequences of our work, none of which we feel must be specifically highlighted here.

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

## A. Detailed Theoretical Analysis

In this section, we provide generalization analysis of our DRRho framework. Given a function class $\mathcal{F}$ and set $\mathcal{X}^n$ with $n$ samples, define

$$N_\infty(\mathcal{F}, \epsilon, n) = \sup_{x \in \mathcal{X}^n} N(\mathcal{F}(x), \epsilon, \|\cdot\|_\infty),$$

where $\mathcal{F}(x) = \{(f(x_1), \ldots, f(x_n)) : f \in \mathcal{F}\}$ for $x \in \mathcal{X}^n$, and

$$N(V, \epsilon, \|\cdot\|) := \inf\{N \in \mathbb{N} : \text{there is an } \epsilon\text{-cover of size } N \text{ of } V \text{ w.r.t. } \|\cdot\|\}.$$

**Theorem A.1** (Theorem 3 in (Duchi & Namkoong, 2016)). *Assume that $f(\cdot) \in [M_0, M_1]$ with $M = M_1 - M_0$ for all $f \in \mathcal{F}$. Let $n \geq 8M^2/t$, $t \geq \log 12$, $\epsilon > 0$, and $\rho \geq 9t$. Then with probability at least $1 - 2(3N_\infty(\mathcal{F}, \epsilon, 2n) + 1)e^{-t}$,*

$$\mathbb{E}[f(X)] \leq \sup_{P:D_\phi(P\|\hat{P}) \leq \frac{\rho}{n}} \mathbb{E}_P[f(X)] + \frac{11}{3} \frac{M\rho}{n} + \left(2 + 4\sqrt{\frac{2t}{n}}\right)\epsilon \tag{13}$$

*for all $f \in \mathcal{F}$. Defining the empirical minimizer*

$$\hat{f} \in \arg\min_{f \in \mathcal{F}} \left\{ \sup_P \left\{ \mathbb{E}_P[f(X)] : D_\phi(P\|\hat{P}) \leq \frac{\rho}{n} \right\} \right\}$$

*we have with the same probability that*

$$\mathbb{E}[\hat{f}] \leq \inf_{f \in \mathcal{F}} \left\{ \mathbb{E}[f] + 2\sqrt{\frac{2\rho}{n} \mathrm{Var}(f)} \right\} + \frac{19M\rho}{3n} + \left(2 + 4\sqrt{\frac{2t}{n}}\right)\epsilon. \tag{14}$$

**Corollary A.2** (Corollary 3.1 in (Duchi & Namkoong, 2016)). *In addition to the conditions of Theorem A.1, let $\mathcal{F}$ have finite VC-dimension $\mathsf{VC}(\mathcal{F})$. Then for a numerical constant $c < \infty$, the bounds in Theorem A.1 hold with probability at least*

$$1 - \left( c\,\mathsf{VC}(\mathcal{F}) \left(\frac{16Mne}{\epsilon}\right)^{\mathsf{VC}(\mathcal{F})-1} + 2 \right) e^{-t}.$$

**Theorem A.3.** *Under the conditions of Theorem A.1, let $d_v$ denote the VC-dimension of $\Theta$ and $c < \infty$ denote a constant.*

1. *For any $\boldsymbol{\theta} \in \Theta$, let $t = \log \frac{1}{\delta} + \log c + d_v \log 16e + 2d_v \log n$ and $\rho \geq 9t$ for $\delta > 0$, then with probability at least $1 - \delta$,*

$$R(\boldsymbol{\theta}) \leq \inf_{\boldsymbol{\theta} \in \Theta} \left( R(\boldsymbol{\theta}) + 2\sqrt{\frac{2\rho}{n} \mathrm{Var}(\ell(\boldsymbol{\theta}, \cdot))} \right) + \left(\frac{25\rho}{3} + 2\right)\frac{M}{n}$$
$$+ \sup_{P:D_\phi(P\|\hat{P}) \leq \frac{\rho}{n}} \mathbb{E}_P[\ell(\boldsymbol{\theta}, \cdot)] - \min_{\boldsymbol{\theta} \in \Theta} \sup_{P:D_\phi(P\|\hat{P}) \leq \frac{\rho}{n}} \mathbb{E}_P[\ell(\boldsymbol{\theta}, \cdot)].$$

2. *For the solution $\boldsymbol{\theta}^*$ of*

$$\min_{\boldsymbol{\theta} \in \Theta} \sup_{P:D_\phi(P\|\hat{P}) \leq \frac{\rho}{n}} \mathbb{E}_P[\ell(\boldsymbol{\theta}, \cdot)], \tag{15}$$

*let $t = \log \frac{1}{\delta} + \log c + d_v \log 16e + 2d_v \log n$ and $\rho \geq 9t$ for $\delta > 0$, then with probability at least $1 - \delta$,*

$$R(\boldsymbol{\theta}^*) \leq \inf_{\boldsymbol{\theta} \in \Theta} \left( R(\boldsymbol{\theta}) + 2\sqrt{\frac{2\rho}{n} \mathrm{Var}(\ell(\boldsymbol{\theta}, \cdot))} \right) + \left(\frac{25\rho}{3} + 2\right)\frac{M}{n}.$$

*Proof.* We first prove the first part of the theorem, which follows the proof of Theorem 3 in (Duchi & Namkoong, 2016). From Theorem A.1 we know for any $f \in \mathcal{F}$, with probability at least $1 - 2(3N_\infty(\mathcal{F}, \epsilon, 2n) + 1)e^{-t}$,

$$\mathbb{E}[f(X)] \leq \sup_{P:D_\phi(P\|\hat{P}) \leq \frac{\rho}{n}} \mathbb{E}_P[f(X)] + \frac{11}{3}\frac{M\rho}{n} + \left(2 + 4\sqrt{\frac{2t}{n}}\right)\epsilon$$

$$= \min_{\hat{f} \in \mathcal{F}} \sup_{P:D_\phi(P\|\hat{P}) \leq \frac{\rho}{n}} \mathbb{E}_P[\hat{f}(X)] + \frac{11}{3}\frac{M\rho}{n} + \left(2 + 4\sqrt{\frac{2t}{n}}\right)\epsilon$$

$$+ \sup_{P:D_\phi(P\|\hat{P}) \leq \frac{\rho}{n}} \mathbb{E}_P[f(X)] - \min_{\hat{f} \in \mathcal{F}} \sup_{P:D_\phi(P\|\hat{P}) \leq \frac{\rho}{n}} \mathbb{E}_P[\hat{f}(X)].$$

Following the proof of Theorem 3 in (Duchi & Namkoong, 2016), setting $\epsilon = \frac{M}{n}$, $t = \log\frac{1}{\delta} + \log c + d_v \log 16e + 2d_v \log n$, we know for any $\boldsymbol{\theta} \in \Theta$, with $\rho \geq 9t$ and probability at least $1 - 2(3N_\infty(\mathcal{F}, \epsilon, 2n) + 1)e^{-t}$,

$$R(\boldsymbol{\theta}) \leq \inf_{\boldsymbol{\theta} \in \Theta} \left(R(\boldsymbol{\theta}) + 2\sqrt{\frac{2\rho}{n}\mathrm{Var}(\ell(\boldsymbol{\theta}, \cdot))}\right) + \left(\frac{25\rho}{3} + 2\right)\frac{M}{n}$$

$$+ \sup_{P:D_\phi(P\|\hat{P}) \leq \frac{\rho}{n}} \mathbb{E}_P[\ell(\boldsymbol{\theta}, \cdot)] - \min_{\boldsymbol{\theta} \in \Theta} \sup_{P:D_\phi(P\|\hat{P}) \leq \frac{\rho}{n}} \mathbb{E}_P[\ell(\boldsymbol{\theta}, \cdot)].$$

From Corollary A.2 we know the probability is at least

$$1 - \left(c\,\mathsf{VC}(\mathcal{F})\left(\frac{16Mne}{\epsilon}\right)^{\mathsf{VC}(\mathcal{F})-1}\right)e^{-t}$$

for some $c < \infty$. Let $d_v$ denote the VC-dimension of $\mathcal{F}$, then we know the probability is at least

$$1 - \left(c\,d_v\left(16n^2e\right)^{d_v-1}\right)\exp(-t)$$

$$= 1 - \exp\left(\log c + \log d_v + (d_v - 1)(\log 16e + 2\log n)\right) \cdot \exp(-t)$$

$$\geq 1 - \exp\left(\log c + d_v \log 16e + 2d_v \log n - t\right)$$

Since $t = \log\frac{1}{\delta} + \log c + d_v \log 16e + 2d_v \log n$, we know that with $\rho \geq 9t$ and probability at least $1 - \delta$,

$$R(\boldsymbol{\theta}) \leq \inf_{\boldsymbol{\theta} \in \Theta} \left(R(\boldsymbol{\theta}) + 2\sqrt{\frac{2\rho}{n}\mathrm{Var}(\ell(\boldsymbol{\theta}, \cdot))}\right) + \left(\frac{25\rho}{3} + 2\right)\frac{M}{n}$$

$$+ \sup_{P:D_\phi(P\|\hat{P}) \leq \frac{\rho}{n}} \mathbb{E}_P[\ell(\boldsymbol{\theta}, \cdot)] - \min_{\boldsymbol{\theta} \in \Theta} \sup_{P:D_\phi(P\|\hat{P}) \leq \frac{\rho}{n}} \mathbb{E}_P[\ell(\boldsymbol{\theta}, \cdot)].$$

This completes the proof for the first part. The second part then follows from the fact that $\boldsymbol{\theta}^*$ is the solution of Equation (15). This completes the proof. $\square$

*Proof of Theorem 4.1.* Since $\tilde{\boldsymbol{\theta}}_*$ minimizes Equation (3), applying Theorem A.3 with $f(\cdot) := \ell(\boldsymbol{\theta}, \cdot) - \ell(\boldsymbol{\theta}_{\mathrm{ref}}, \cdot)$ (where the range becomes $2M$), letting $t = \log\frac{1}{\delta} + \log c + d_v \log 16e + 2d_v \log n$ and $\rho \geq 9t$ for given $\delta > 0$ and constant $c < \infty$, then with probability at least $1 - \delta$,

$$R(\tilde{\boldsymbol{\theta}}_*) - R(\boldsymbol{\theta}_{\mathrm{ref}}) \leq \inf_{\boldsymbol{\theta} \in \Theta_o} \left(R(\boldsymbol{\theta}) - R(\boldsymbol{\theta}_{\mathrm{ref}}) + \sqrt{\frac{2\rho}{n}\mathrm{Var}(\ell(\boldsymbol{\theta}, \cdot) - \ell(\boldsymbol{\theta}_{\mathrm{ref}}, \cdot))}\right) + \left(\frac{50\rho}{3} + 4\right)\frac{M}{n},$$

which is

$$R(\tilde{\boldsymbol{\theta}}_*) \leq \inf_{\boldsymbol{\theta} \in \Theta_o} \left(R(\boldsymbol{\theta}) + \sqrt{\frac{2\rho}{n}\mathrm{Var}(\ell(\boldsymbol{\theta}, \cdot) - \ell(\boldsymbol{\theta}_{\mathrm{ref}}, \cdot))}\right) + \left(\frac{50\rho}{3} + 4\right)\frac{M}{n}.$$

This completes the proof. $\square$

*Proof of Corollary 4.2.* From the definition we have $\boldsymbol{\theta}_* \in \Theta$. Thus we know

$$\inf_{\boldsymbol{\theta} \in \Theta_o} \left( R(\boldsymbol{\theta}) + \sqrt{\frac{2\rho}{n} \mathrm{Var}(\ell(\boldsymbol{\theta}, \cdot) - \ell(\boldsymbol{\theta}_{\mathrm{ref}}, \cdot))} \right) \leq R(\boldsymbol{\theta}_*) + \sqrt{\frac{2\rho}{n} \mathrm{Var}(\ell(\boldsymbol{\theta}_*, \cdot) - \ell(\boldsymbol{\theta}_{\mathrm{ref}}, \cdot))}.$$

Plugging the above inequality into Theorem 4.1, we get

$$R(\tilde{\boldsymbol{\theta}}_*) \leq R(\boldsymbol{\theta}_*) + \sqrt{\frac{2\rho}{n} \mathrm{Var}(\ell(\boldsymbol{\theta}_*, \cdot) - \ell(\boldsymbol{\theta}_{\mathrm{ref}}, \cdot))} + \frac{C_2}{n}.$$

This completes the proof. $\square$

*Proof of Corollary 4.3.* Since $\boldsymbol{\theta}_{\mathrm{ref}} \in \Theta$, we know

$$\inf_{\boldsymbol{\theta} \in \Theta_o} \left( R(\boldsymbol{\theta}) + \sqrt{\frac{2\rho}{n} \mathrm{Var}(\ell(\boldsymbol{\theta}, \cdot) - \ell(\boldsymbol{\theta}_{\mathrm{ref}}, \cdot))} \right) \leq R(\boldsymbol{\theta}_{\mathrm{ref}}) + \sqrt{\frac{2\rho}{n} \mathrm{Var}(\ell(\boldsymbol{\theta}_{\mathrm{ref}}, \cdot) - \ell(\boldsymbol{\theta}_{\mathrm{ref}}, \cdot))} = R(\boldsymbol{\theta}_{\mathrm{ref}}).$$

Plugging the above inequality into Theorem 4.1, and subtracting $R(\boldsymbol{\theta}_*)$ on both sides, we get

$$R(\tilde{\boldsymbol{\theta}}_*) - R(\boldsymbol{\theta}_*) \leq R(\boldsymbol{\theta}_{\mathrm{ref}}) - R(\boldsymbol{\theta}_*) + \frac{C_2}{n}.$$

This completes the proof. $\square$

## B. Connection between DRRho and Existing Methods

In Section 4 we show that our proposed framework can be applied to data selection, weighting and sampling. Here we provide a comparison between our framework and existing methods.

**Data Selection**: With the CVaR divergence, the DRRho objective becomes the average of top-$k$ RHO losses (Equation (4)). This is connected to RHO (Mindermann et al., 2022) and RHO-1 (Lin et al., 2024). The key idea of both RHO and RHO-1 is both to first sample a large batch of data points, then leverage a reference model to compute the RHO loss for each sample in the large batch, and in the end only samples with the largest RHO loss values are kept for back-propagation. The major difference between our framework and the two existing works is that our objective will select top samples in the *whole dataset* for back-propagation, while existing works operate in batch level.

**Data Weighting / Sampling**: With the KL-divergence, our framework becomes a data weighting method that assigns different weights to different samples. In the CLIP training setting, data weighting is applied to every combination of image and text. Evans et al. (2025) applied the idea for data sampling and proposed ActiveCLIP. Specifically, given a large batch of data points, they first compute the RHO loss using a reference model for each sample in the batch, then they pass the loss scores through a softmax function and view it as a probability distribution to sample a small batch for back-propagation. Evans et al. (2024a) builds upon the idea of ActiveCLIP and proposed JEST, which leverages the RHO loss as well but use a fine-grained sampling approach due to the special structure of the contrastive loss. Note that both ActiveCLIP and JEST are proposed for training CLIP models, which is the same as our DRRho-CLIP. The core difference between our DRRho-CLIP and ActiveCLIP and JEST is that their losses are used for data selection of anchor data while ours has an effect of data re-weighting of the negative data for each anchor data. Moreover, their methods use mini-batch contrastive loss to define a RHO loss. While we define the loss using all negative data in the training dataset (instead of the mini-batch) and use a rigorous optimization approach (SogCLR) to optimize the objective.

## C. More Experiment Results

### C.1. Implementation Details

Our implementation is based on FastCLIP (Wei et al., 2024), which includes the implementation of FastCLIP and OpenCLIP. We leveraged the code provided by Evans et al. (2024a) with minor modification for our implementation of JEST and JEST (Top-k). We merge the code released by Vasu et al. (2024) into our code base for the implementation of MobileCLIP.

## C.2. Hyperparameters

**DRRho-CLIP**. We set the learning rate to 3.125e-4 for training ViT-B/16, and set the learning rate to 8e-4 for training ViT-B/32. In most experiments in Section 6, we set the temperature to 0.01. The only exception is the one with reference model trained on DFN-9M where the temperature is learnable (cf. Appendix C.3) and the initial temperature is set to 0.07.

**Baselines: OpenCLIP and FastCLIP** We use the FastCLIP-v3 from Wei et al. (2024) as the FastCLIP baseline. For both OpenCLIP and FastCLIP, we set the learning rate to 3.125e-4 for training ViT-B/16, and set the learning rate to 8e-4 for training ViT-B/32. We set the initial temperature to 0.07. For FastCLIP, we set the learning rate of $\tau$ to be 1/4 of the learning rate of the model, we set $\rho$ to 11.0 for training ViT-B/16 and 8.5 for training ViT-B/32.

**JEST and JEST (Top-$k$)** We set the learning rate to 3.125e-4 for training ViT-B/16. Following Evans et al. (2024a), the selection ratio is set to 0.2, which means only 20% of the super-batch will be used for training. The size of the super-batch at each iteration is 25600 and the size of the mini-batch for training is 5120, which is the same as other methods. We set the number of chunks to 2 so that the chunk size of 2560, which is close to the value 2048 suggested by Evans et al. (2024a). At selection ratio 0.2, JEST and JEST (Top-$k$) spends 5 times more compute on forward propagation (and the same amount of compute on backward propagation) compared with other approaches. Since the amount of compute of forward propagation is approximately 1/3 of that of backward propagation (Evans et al., 2024a), we increase the number of iterations to 1.87 times of that of other approaches so that these methods consume the same amount of compute.

**MobileCLIP, FastCLIP with Distillation, DRRho-CLIP with Distillation** We set the learning rate to 3.125e-4 for training ViT-B/16. We set the value of $\lambda$ of DRRho-CLIP with Distillation to be the same as that of FastCLIP with Distillation. On DFN-12M, for MobileCLIP, we set the value of $\lambda$ to 0.4; for FastCLIP with Distillation, we set the value of $\lambda$ to 0.25. On DFN-192M, for MobileCLIP, we set the value of $\lambda$ to 0.75; for FastCLIP with Distillation, we set the value of $\lambda$ to 0.25. On DFN-192M with reference model ViT-B/32 (WIT-400M), since the performance of the reference model is low, the value $\lambda$ is set to 0.0 (i.e., no distillation) for all three methods.

## C.3. Additional Ablation Study

**Comparison between Fixed and Learnable Temperature $\tau$.** In most our experiments of DRRho-CLIP, we set the temperature $\tau$ to 0.01 and fix it throughout training. The only exception is the one with reference model trained on DFN-9M where we set $\tau$ to a learnable hyperparameter as in FastCLIP(-v3), which gives better performance than using a fixed temperature. To further study this, we conduct experiments to compare the performance of learnable temperature and fixed temperature. Mathematically, the loss formulation with learnable temperature is similar to Equations (8) and (9) except that the temperature now becomes a parameter that can be optimized:

$$\min_{\boldsymbol{\theta},\tau} \frac{1}{n} \sum_{i=1}^{n} \tau \log \left( \frac{1}{n} \sum_{j=1}^{n} \exp(\frac{\hat{\ell}(\boldsymbol{\theta}, \boldsymbol{\theta}_{\text{ref}}, \boldsymbol{x}_i, \boldsymbol{y}_j)}{\tau}) \right) + \tau \log \left( \frac{1}{n} \sum_{j=1}^{n} \exp(\frac{\hat{\ell}(\boldsymbol{\theta}, \boldsymbol{\theta}_{\text{ref}}, \boldsymbol{y}_i, \boldsymbol{x}_j)}{\tau}) \right) + 2\tau\rho,$$

where $\rho > 0$ is a hyperparameter. We conduct experiments of DRRho-CLIP with fixed and learnable temperature on different target model and reference model combinations. We present the results in Table 2 and Figure 5. We find that, for a reference model with high performance, using a fixed temperature leads to slightly better performance than using a learnable temperature. While for reference models with low performance, a learnable temperature yields higher performance.

Table 2: ImageNet Top 1 accuracy of DRRho-CLIP with fixed and learnable temperature on different target models and reference models.

| Target Model (Data) | Reference Model (Data) | ImageNet Top 1 Accuracy | | | |
| | | FastCLIP | DRRho-CLIP (Learnable $\tau$) | DRRho-CLIP (Fixed $\tau$) | Reference |
| --- | --- | --- | --- | --- | --- |
| ViT-B/16 (DFN-12M) | ViT-B/32 (DFN-12M) | 43.49 | **46.31** | 37.65 | 36.27 |
| ViT-B/16 (DFN-12M) | ViT-B/32 (WIT-400M) | 43.49 | 49.81 | **49.91** | 63.32 |
| ViT-B/16 (DFN-192M) | ViT-B/32 (WIT-400M) | 67.37 | 68.17 | **68.84** | 63.32 |

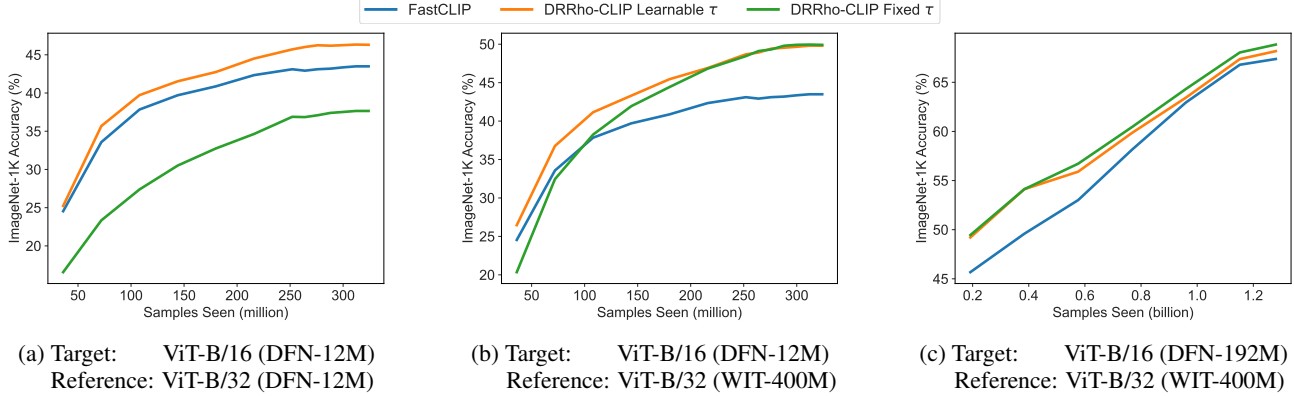

(a) Target:     ViT-B/16 (DFN-12M)
Reference: ViT-B/32 (DFN-12M)

(b) Target:     ViT-B/16 (DFN-12M)
Reference: ViT-B/32 (WIT-400M)

(c) Target:     ViT-B/16 (DFN-192M)
Reference: ViT-B/32 (WIT-400M)

Figure 5: ImageNet Top 1 accuracy curves of DRRho-CLIP with fixed and learnable temperature on different target models and reference models.

### C.4. Comparison with Baselines

The distillation loss we consider is the same as in Vasu et al. (2024), and is defined as follows

$$
\begin{aligned}
\mathcal{L}_{\text{distill}}(\boldsymbol{\theta}, \boldsymbol{\theta}_{\text{ref}}, \mathcal{B}) := & - \frac{1}{|\mathcal{B}|^2} \sum_{i \in \mathcal{B}} \sum_{j \in \mathcal{B}} \frac{\exp(\hat{s}(\boldsymbol{x}_i, \boldsymbol{y}_j)/\hat{\tau})}{\sum_{k \in \mathcal{B}} \exp(\hat{s}(\boldsymbol{x}_i, \boldsymbol{y}_k)/\hat{\tau})} \log \frac{\exp(s(\boldsymbol{x}_i, \boldsymbol{y}_j)/\tau)}{\sum_{k \in \mathcal{B}} \exp(s(\boldsymbol{x}_i, \boldsymbol{y}_k)/\tau)} \\
& - \frac{1}{|\mathcal{B}|^2} \sum_{i \in \mathcal{B}} \sum_{j \in \mathcal{B}} \frac{\exp(\hat{s}(\boldsymbol{x}_j, \boldsymbol{y}_i)/\hat{\tau})}{\sum_{k \in \mathcal{B}} \exp(\hat{s}(\boldsymbol{x}_k, \boldsymbol{y}_i)/\hat{\tau})} \log \frac{\exp(s(\boldsymbol{x}_j, \boldsymbol{y}_i)/\tau)}{\sum_{k \in \mathcal{B}} \exp(s(\boldsymbol{x}_k, \boldsymbol{y}_i)/\tau)},
\end{aligned}
\tag{16}
$$

where $s(\boldsymbol{x}_i, \boldsymbol{y}_j)$ ($\hat{s}(\boldsymbol{x}_i, \boldsymbol{y}_j)$, resp.) denotes the cosine similarity between $i$-th image and $j$-th text output by the target model (reference model, resp.).

In Table 3, we present the results of different baselines on DFN-12M with 320M samples seen.

In the following figure, we plot the ImageNet-1K Top 1 accuracy of different models trained using OpenCLIP or DRRho-CLIP.

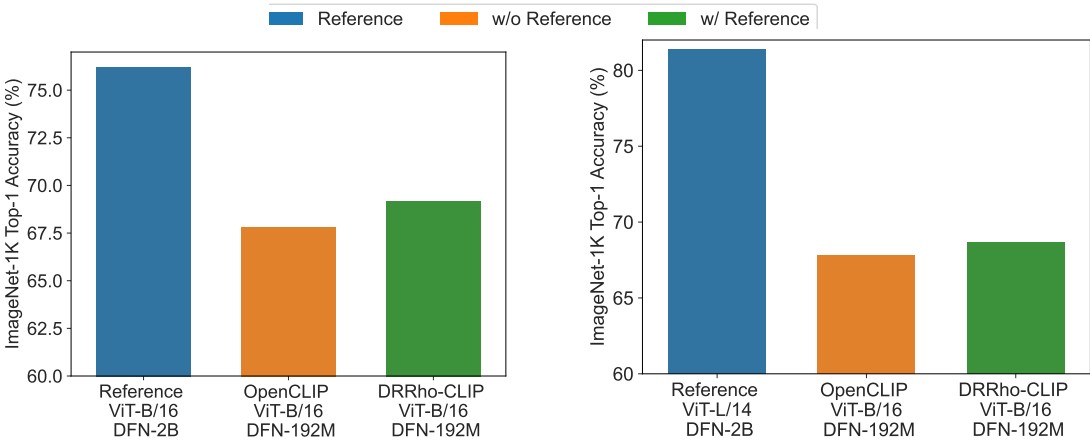

Figure 6: Zero-shot Top 1 Accuracy on ImageNet-1K of different models. Reference model is a ViT-B/16 (left figure) or ViT-L/14 (right figure) pretrained on DFN-2B dataset with 12.8B samples seen and a batch size of 90112 (Fang et al., 2024); OpenCLIP model is a ViT-B/16 trained on DFN-192M dataset with 1.28B samples seen and a batch size of 8192 (Fang et al., 2024), DRRho-CLIP model was trained on DFN-192M dataset with 1.28B samples seen and using the reference model with a batch size of 5120. The latter two use the same model structure.

Table 3: Comparison of different methods on DFN-12M with 320M samples seen. Reference denotes the performance of the reference model. OpenCLIP and FastCLIP does not leverage a reference model. For distillation-based methods, the reference model is also the teacher model for distillation.

| Metric | Method | | Reference Model | |
|---|---|---|---|---|
| | | ViT-B/32 (WIT-400M) | ViT-B/16 (DFN-2B) | ViT-L/14 (DFN-2B) |
| ImageNet Top 1 | Reference | 63.32 | 76.23 | 81.41 |
| | OpenCLIP | 41.10 | 41.10 | 41.10 |
| | FastCLIP | 43.49 | 43.49 | 43.49 |
| | JEST | 36.78 | 34.63 | 33.46 |
| | JEST (Top-k) | 36.30 | 34.75 | 33.38 |
| | DRRho-CLIP | **49.95** | **47.57** | **46.70** |
| | MobileCLIP (w/ Distillation) | 52.77 | 48.01 | 45.80 |
| | FastCLIP (w/ Distillation) | 53.21 | 50.38 | 47.86 |
| | DRRho-CLIP (w/ Distillation) | **53.33** | **52.58** | **49.84** |
| Datacomp | Reference | 52.27 | 60.75 | 66.65 |
| | OpenCLIP | 37.67 | 37.67 | 37.67 |
| | FastCLIP | 38.57 | 38.57 | 38.57 |
| | JEST | 35.59 | 35.42 | 34.36 |
| | JEST (Top-k) | 34.60 | 34.56 | 34.88 |
| | DRRho-CLIP | **43.02** | **42.15** | **41.47** |
| | MobileCLIP | 44.57 | 41.68 | 39.81 |
| | FastCLIP (w/ Distillation) | 44.57 | 43.72 | 42.04 |
| | DRRho-CLIP (w/ Distillation) | **46.29** | **43.88** | **42.49** |

## C.5. Scaling Law

In Figure 7 we plot the relationship $\log E = \log \alpha + \beta \log C$ for DRRho-CLIP and OpenCLIP, where $E$ denotes the ImageNet error rate and $C$ denotes the combined amount of compute used by the target model and the reference model. The main difference between Figure 7 and Figure 2 is that the former considers the compute used by the reference model while the latter does not. Similar to Figure 2, we also observer that DRRho-CLIP has a better scaling law than OpenCLIP with smaller $\beta$.

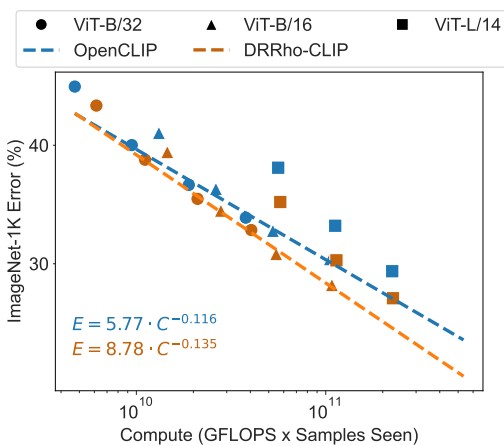

Figure 7: Scaling performance of OpenCLIP (Cherti et al., 2023) and the proposed DRRho-CLIP, which uses the OpenAI CLIP model (Radford et al., 2021) as the reference model. We conduct experiments of the two methods under different settings to fit scaling laws, as shown in the bottom left corner (c.f. Section 6 for more detail).

