# OpenReview forum: "Model Steering: Learning with a Reference Model Improves Generalization Bounds and Scaling Laws"
_ICML.cc/2025/Conference — ICML 2025 spotlightposter_

### Official Review · Reviewer_26QA · 2025-02-24

**Overall Recommendation:** 3

**Summary:**

This paper theoretically studies the mechanism behind a learning paradigm, called Learning with a Reference Model (LEAR) and proposes a new learning algorithm that achieves better scaling than the naive approach. They first relate the RHO loss with DRO and show how using the RHO loss can improve the generalizability of DRO. They then show that when a certain divergence function is used, RHO+DRO is equivalent to weighting samples with RHO loss, which theoretically explains why LEAR is useful. To optimize RHO+DRO, they adopt the SogCLR algorithm. They show that their approach has better sample efficiency than the naive method and the distillation methods.

**Claims And Evidence:**

1. The author claims that RHO can achieve a smaller variance, and verify it empirically.
2. The author claims that RHO+DRO can achieve a better sample complexity/scaling law. It is also verified.

**Essential References Not Discussed:**

Not aware of

**Experimental Designs Or Analyses:**

No crucial problems found.

**Methods And Evaluation Criteria:**

They evaluate the algorithm by training CLIP models and evaluate ImageNet, which seems reasonable. However, a caveat is that they only experiment with this single task.

**Other Comments Or Suggestions:**

If I understood correctly, this paper has two goals, one is to provide the theoretical foundation of LEAR, which, in this paper, seem to be specific to (Evans et al., 2024a;b). The other one is to propose a better algorithm for LEAR. While I don't see apparent problems for the second goal, the one seems to be done very implicitly.

I would suggest that the authors provide some background about the specific LEAR methods this paper can provide insights into. This would make the purpose of Section 4 clearer.

Beside, while it's the convention to call scaling law scaling law, the word "law" is actually misleading, as it is not a real law. If possible, I would suggest that you just say you scale better.

**Other Strengths And Weaknesses:**

Strength

- The authors show that their approach indeed has great sample efficiency compared with the naive and the distillation approaches.

Weakness

- Based on what I understood, the proposed method optimizes the same objective as (Evans et al., 2024a;b). Comparing with their approach seems important.
- Though in the introduction, the authors motivate this paper by the lack of theoretical understanding of LEAR, this paper doesn't seem to be able to explain ALL LEAR methods. It should be more specific about *which* LEAR method(s) this paper is for.
- The flow of this paper could be improved. See suggestions.
- Please also see questions.

In sum, while I acknowledge this paper has its technical merits, I suppose this paper was written in a rush. Thus, while technically I suggest this paper is probably acceptable, I also suggest that another round of revision will make it better and more up to the level of ICML.

Please let me know if I misunderstood anything.

**Questions For Authors:**

1. One thing I didn't understand is what the x-axis of figure 1 (Samples Seen (million)) means. Does it represent the training time? If not, why did you use different line to represent different ratio of data used? And why is the trend of Figure 2 different from Figure 3? It is confusing because these two figures have the same x and y axes.
2. Would this method works for other tasks? Is there a reason to focus only on CLIP?

**Relation To Broader Scientific Literature:**

Based on the paper, this work is related to DRO papers and provides the theoretical foundation of [1]

[1] Data curation via joint example selection further accelerates multimodal learning

**Theoretical Claims:**

They don't seem to be incorrect.

A caveat is that there are too many constants and it's hard to understand their meaning. As a result, I am not sure whether Corollary 4.3 is correct or non-trivial.

---

> ### Author Rebuttal · Authors · 2025-04-01
>
> We thank the reviewer for appreciating the technical merits of this paper. We will follow the reviewer's suggestions to improve the paper, which should be an easy task.
>
> > **Q1**: Would this method work for other tasks? Is there a reason to focus only on CLIP?
>
> **A**: Yes! We have discussed that the existing works of employing the RHO loss for data selection or sampling for different tasks can be considered as heuristic implementation of the proposed framework (Section 4). These different tasks include classification (Mindermann et al., 2022) and training LLMs (Lin et al. 2024). We have discussed the reason for our focus on CLIP at the beginning of Section 5.
>
> > **Q2**: How $C_2$ on the right hand side at line 733 is derived in the proof of Corollary 4.3.
>
> **A**: The constant $C_2$ in Corollary 4.3 is the same as that in Theorem 4.1. In Theorem 4.1, we derived $$R(\tilde{\theta}_*)\leq \underbrace{\inf _{\theta\in \Theta} \left(R(\theta) + \sqrt{\frac{2\rho}{n} \mathrm{Var}(\ell(\theta, \cdot)- \ell(\theta _{\mathrm{ref}}, \cdot))}\right)} _{\mathrm{term1}}+ \frac{C _{2}}{n},$$ where $C _2=(50\rho/3+4)M$. In the proof of Corollary 4.3, we showed that term1 in the brace is upper bounded by $R(\theta _\mathrm{ref})$. Hence, $C_2$ is the same as in Theorem 4.1.
>
> > **Q3**: Does the proposed method optimize the same objective as (Evans et al., 2024a;b)? Comparing with their approach seems important.
>
> **A**: We would like to point out that we indeed compared with Evans et al., 2024a (JEST). We omitted the comparison with Evans et al., 2024b (ActiveCLIP) since JEST is a later work and improved over it by the same groups of authors. Our method does not optimize the same objective as their methods:
> - their methods use mini-batch contrastive loss to define a RHO loss. While we define the loss using all negative data in the training dataset (instead of the mini-batch) and use a rigorous optimization approach (SogCLR) to optimize the objective.
> - their loss is used for data selection of anchor data, i.e., selecting subset for training. In contrast, our DRRho contrastive loss, defined by leveraging the relationship between global contrastive loss and DRO, has an effect of data re-weighting of the negative data for each anchor data.
>
> Moreover, their methods consume more resources for data selection due to sampling from a larger batch. We compared the performance of JEST and DRRho-CLIP on DFN-12M and DFN-192M with fixed amount of compute. The results (c.f. Tables 1 and 3 in the paper) showed that DRRho-CLIP significantly outperforms JEST.
>
> > **Q4**: This paper doesn't seem to be able to explain all LEAR methods. It should be more specific about which LEAR method(s) this paper is for.
>
> **A**: LEAR refers to, as stated at the beginning of the abstract, *leveraging a pretrained model ... through strategic data selection or weighting*. Thus we focus on methods for data selection and weighting. We categorize existing works of leveraging a pretrained model into different families in Section 2, where in Lines 120-140 we provide background about the specific LEAR methods that motivates this work, such as RHO (Mindermann et al., 2022), RHO-1 (Lin et al., 2024), ActiveCLIP (Evans et al., 2024b) and JEST (Evans et al., 2024a) to make Section 4 clearer.
>
> > **Q5**: I would suggest that the authors provide some background about the specific LEAR methods this paper can provide insights into. This would make the purpose of Section 4 clearer.
>
> **A**: Thank you for the suggestion. We will give more background about the related LEAR methods mentioned in the above question to make Section 4 clearer.
>
> > **Q6**: While it's the convention to call scaling law scaling law, the word "law" is actually misleading, as it is not a real law. If possible, I would suggest that you just say you scale better.
>
> **A**: While we agree with the reviewer that the term "law" is misleading or sometimes overclaims, however, we hesitate to invent a new term as it has been widely accepted in the literature. Cherti et al. (2023) also used the term "scaling law" for CLIP training.
>
> > **Q7**: Does the x-axis of Figure 2 represent the training time? If not, why did you use different line to represent different ratio of data used?
>
> **A**: The x-axis in Figure 2 represents the number of samples seen, which is proportional to training time. Different ratio of data means the training dataset size is different. For example, 100% data means the whole dataset is used during training, while 50% data means only half of the dataset is used during training. For different datasets, we set the total number of samples seen to be the same, which means on smaller datasets there will be more epochs.
>
> > **Q8**: Why is the trend of Figure 2 different from Figure 3?
>
> **A**: In Figure 2, we use ImageNet-1K Top 1 Accuracy as the y-axis. While in Figure 3, we use ImageNet-1K Error as the y-axis, which is equal to 100% - ImageNet-1K Top 1 Accuracy. Thus the two figures have different trends.

---

> > ### Comment · Reviewer_26QA · 2025-04-03
> >
> > Thanks for the clarification. I wish I have time to check the proof again. Before I have time to check the proof and really find a problem (if exists), I think I have no major reasons to reject this paper, so I will increase my score for now.

---

> > > ### Author Response · Authors · 2025-04-07
> > >
> > > We thank the reviewer for their valuable suggestions and for raising the score after rebuttal.

---

### Official Review · Reviewer_Mms9 · 2025-03-13

**Overall Recommendation:** 4

**Summary:**

The paper establishes a theoretical framework for RHO-based learning with a reference model using DRO as the perspective and introduces a novel DRRho risk. It further applies DRRho-based LEAR to CLIP, achieving good and data-efficient performance.

## update after rebuttal
My overall evaluation remains unchanged.

**Claims And Evidence:**

The authors make several key claims:
- The DRRho framework improves generalization via variance reduction.
- DRRho-CLIP outperforms heuristic methods.
- DRRho-CLIP is more data-efficient than vanilla ERM.

Overall, the claims in the paper are well supported.

**Essential References Not Discussed:**

I didn't notice any missing key references.

**Experimental Designs Or Analyses:**

The experiments are well-designed and carefully analyzed to support their theoretical claims and the efficiency of their method.

**Methods And Evaluation Criteria:**

The DRO-based DRRho risk and its application to CLIP are well-motivated.

**Other Comments Or Suggestions:**

No other comments.

**Other Strengths And Weaknesses:**

**Strengths**:
- The paper is well-written and easy to follow.
- The theoretical justification of RHO-based LEAR using DRO is creative and reasonable.
- The analysis is comprehensive, supported by sufficient theoretical and experimental evidence.

**Weaknesses**:
The paper offers a potential theoretical explanation via generalization bounds for why LEAR is more data-efficient than ERM, and it is expected that a better reference model should lead to greater improvements in target model training (as shown in Corollaries 4.2 and 4.3). However, we can observe in Table 1 that a more powerful reference model does not always yield superior target model performance.
It would be beneficial if the authors could clarify and explain this discrepancy.

**Questions For Authors:**

I have a question regarding the interpretation of Corollary 4.3. The paper claims that "DRRho needs only $n = O( \sqrt{ m })$ samples, which dramatically reduces the sample complexity $O(m)$ of ERM without a reference model." However, when viewed through the lens of Theorem 4.1 or Corollary 4.2, it seems that the sample complexity would still be $O(m)$. Could the authors clarify this apparent contradiction in the sample complexity analysis?

**Relation To Broader Scientific Literature:**

The paper builds on DRO (Duchi & Namkoong, 2016), contrastive learning (Qiu et al., 2023), and LEAR (Mindermann et al., 2022).

**Theoretical Claims:**

I didn’t check the detailed proofs in the Appendix.

---

> ### Author Rebuttal · Authors · 2025-04-01
>
> We thank the reviewer for their valuable comments and suggestions.
>
> > **Q1**: More powerful reference model does not always yield superior target model performance. But Corollaries 4.2 and 4.3 show that a better reference model should lead to greater improvements in target model training.
>
> **A**: There is misunderstanding of results in Corollaries 4.2 and 4.3 regarding that more powerful reference model yields superior target model performance. Corollary 4.2 shows that the generalization depends on the variance of the RHO loss for a reference model. However, a powerful reference model does not necessarily have a small variance of the RHO loss, i.e., $\mathrm{Var}(\ell(\theta_*, \cdot) - \ell(\theta_\mathrm{ref}, \cdot))$, where $\theta_*$ is the optimal solution in the considered model space. Corollary 4.3 only compares a reference model in **the same space** of the target model. However, in Table 1 except for ViT-B/16, other two reference models (ViT-B/32, ViT-L/14) are not in the same model space of the target model ViT-B/32. Hence, we cannot apply Corollary 4.3.
>
> We would like to point out that this phenomenon is also empirically observed in other works that leverage a reference model, e.g. DFN (Fang et al., 2024), JEST (Evans et al. 2024a) and MobileCLIP (Vasu et al., 2024), where stronger reference models did not necessarily lead to more powerful target models.
>
> To better verify Corollary 4.3,  we have conducted the following experiments with ViT-B/16 as target model and ViT-B/16 trained on different subsets of DFN-12M as reference models. We list the ImageNet-1K Top 1 Accuracy of both the target model and the reference model in the following table, from which we can observe that the performance of the target model and the reference model is positively correlated.
>
> | Target Model (Data, Samples Seen) | Reference Model | Target Model Performance | Reference Model Performance |
> | -- | -- | -- | -- |
> | ViT-B/16 (DFN-12M, 320M) | ViT-B/16 (DFN-6M, 320M) | 42.50 | 30.19 |
> | ViT-B/16 (DFN-12M, 320M) | ViT-B/16 (DFN-9M, 320M) | 46.80 | 39.09 |
> | ViT-B/16 (DFN-12M, 320M) | ViT-B/16 (DFN-12M, 320M) | 48.88 | 43.49 |
>
>
> > **Q2**: Regarding interpretation of Corollary 4.3 and do Theorem 4.1 or Corollary 4.2 still indicate $n=O(m)$ sample complexity?
>
> **A**: The interpretation of Corollary 4.3 about the reduced sample complexity is to guarantee that the generalization error of the learned model $R(\tilde\theta_*) - R(\theta_*)$ by our framework is on par with that of the reference model $R(\theta_{\text{ref}}) - R(\theta_*)$, where $m$ is the data size for training the reference model. If we want to use Corollary 4.2 for deriving $R(\tilde\theta_*) - R(\theta_*)$ in the same order $1/\sqrt{m}$ of $R(\theta_{\text{ref}}) - R(\theta_*)$ , it will imply that $n = \max(\sqrt{m}, m\mathrm{Var}(\ell(\theta_*, \cdot) - \ell(\theta_\mathrm{ref}, \cdot)))$. It could be still much better than $O(m)$ as $\mathrm{Var}(\ell(\theta_*, \cdot) - \ell(\theta_\mathrm{ref}, \cdot))$ could be very small. In this context, we cannot ignore $\mathrm{Var}(\ell(\theta_*, \cdot) - \ell(\theta_\mathrm{ref}, \cdot))$. Hence, it is more convenient to use Corollary 4.3 to make this comparison argument. Thus the two results do not contradict to each other.

---

### Official Review · Reviewer_DaNa · 2025-03-14

**Overall Recommendation:** 4

**Summary:**

The paper proposed DRRho risk minimization with a reference model and provided a theoretical analysis of it. It also applied this approach to training the CLIP model. Experiments show that the proposed method achieves better performance than the baselines.

**Claims And Evidence:**

The claims made in the submission are supported by clear and convincing evidence.

**Essential References Not Discussed:**

I didn’t notice any.

**Experimental Designs Or Analyses:**

The experimental design is valid and demonstrates the effectiveness of the proposed method.

**Methods And Evaluation Criteria:**

The datasets and benchmarks are appropriate and relevant to the problem. However, since I am not an expert on this topic, I’ll leave it to other reviewers to judge whether the baselines are comprehensive and the comparisons are sufficient.

**Other Comments Or Suggestions:**

I don’t have other comments.

**Other Strengths And Weaknesses:**

Overall, I find the contribution solid, with both theoretical insights and experiments showing improvement over the baseline. However, I am not entirely sure about the comprehensiveness of the evaluation and comparison, so I would like to hear other reviewers’ opinions.

**Questions For Authors:**

I don’t have other questions.

**Relation To Broader Scientific Literature:**

The paper is related to the literature on distributionally robust optimization. Although the technique itself is well-established, combining it with a reference model and providing a theoretical analysis appears to be a novel contribution. Furthermore, applying it to training CLIP, a relatively large-scale problem with practical significance, seems like a solid contribution.

**Theoretical Claims:**

I didn’t go through all the proofs in detail, but the theoretical results seem convincing to me.

---

> ### Author Rebuttal · Authors · 2025-04-01
>
> We thank the reviewer for their constructive and positive evaluation of our work. We are happy to address any concerns the reviewer may have in later stage.

---

### Official Review · Reviewer_uJqj · 2025-03-25

**Overall Recommendation:** 3

**Summary:**

Authors present a framework for using available open weights model to improve model training on given dataset (learning with a reference model - LEAR). The framework is based on distributionally robust optimization (DRO). DRO makes use of available data empirical distribution to create perturbed data distributions and use those for worst case risk minimization. Author further employ RHO loss, a generic loss aiming on identifying data points worth learning, to obtain a risk function via applying DRO (DRRHO risk)  They derive theoretical generalization bounds using the risk, aiming to explain how DRRHO improves generalization. They study their method on example of CLIP training, using pre-trained CLIP models as reference. Obtained DRRho CLIP (using various reference models) is compared to various baselines, stating its advantages in downstream task performance and data efficiency.

**Claims And Evidence:**

To test their claims of enhancing generalization via training with reference model, authors perform CLIP training guided by various reference models on various scales and measure model performance via various well-established benchmarks. I think authors' approach is valid and evidence they gather is sufficient to argue for their procedure being useful. Scaling law derivation claim seems overblown. Authors present for single fixed model scale (ViT B/16) measurements across 4 different samples seen scales, through which they fit a power law. This is not what is known as full scaling law. Eg cited work by Cherti et al performs measurements on combinations of 5 different model scales, 3 different samples seen scales (3B, 12B and 34B) and 3 different data scales (80M, 400M and 2B), each of which is tuned to construct a Pareto front corresponding to minimum error on downstream tasks. This full scaling law derivation is not what authors perform and thus only limited (if any) conclusions can be drawn from the Fig 3.

**Essential References Not Discussed:**

Authors cite relevant works properly.

**Experimental Designs Or Analyses:**

Experimental design of CLIP training with reference model seems sound. The derivation of scaling law is not executed properly, or stated differently, there is no scaling law presented in the study although the title strongly suggests so.

**Methods And Evaluation Criteria:**

Authors derived theoretical generalization bounds which they use to backup various heuristics for data filtering / selection that lead to training of models with stronger generalization. They enhance standard contrastive loss of CLIP with DRO loss component that includes reference model, conducting training of DRRHO-CLIP. Datasets used are well established CC12M and DFN subsets (DFN-9M and DFN-128M). Evaluation used well established CLIP benchmarks, eg those used in DataComp, which are used to compare DRRHO-CLIP with other strong reference baselines. Methods and evaluation make sense.

**Other Comments Or Suggestions:**

Text is well written and is easy to follow.

UPDATE: raising score to 3 after scaling law extension experiments by the authors.

**Other Strengths And Weaknesses:**

Strength of the paper is in establishing theoretical grounds for learning with reference model and testing the approach on important scenario of CLIP training, which often uses distillation as technique to boost model performance. Weakness is the missing scaling law that is announced in the paper title. Authors do not derive proper full scaling law, picking only one fixed model scale (ViT B/16). Based on this plot, authors attempt to make a conclusion how DRRHO CLIP compares to openCLIP, which does not work, as for such comparison full scaling law would have been necessary. Authors can only conclude that for ViT B/16 and rather small span of samples seen scales, DRRHO CLIP has advantage over openCLIP without reference model. It is not clear what happens across scales and on larger scales , eg L/14 , H/14, from this examination (full scaling law would allow prediction for those larger scales). Moreover, the compute necessary for using reference model (which has also to compute loss on the training data during training) is not incorporated into considerations for model comparison. In general thus, using total compute on x-axis for learning procedure comparison via scaling law derivation should have been the correct approach here if aiming for strong conclusions about advantage of the learning procedure over other baselines.

**Questions For Authors:**

Scaling law derivation on smaller scales used in the work should have been possible, as those experiments are not expensive. Why was the derivation of full scaling laws (eg following Cherti et all 2023) was ommitted in the work? It would be also good to plot FLOPs vs performance, accounting for FLOPs used by reference model - can authors provide such a plot?

**Relation To Broader Scientific Literature:**

The work fits well into landscape of language-vision learning research and distillation methods.

**Theoretical Claims:**

Theoretical generalization bounds obtained by authors seem correct, as well as introduction of the DRO loss into CLIP training.

---

> ### Author Rebuttal · Authors · 2025-04-01
>
> We thank the reviewer for the suggestion on experiments. We believe that the comments raised by the reviewer are not critical drawbacks of this paper. We request the reviewer to consider our contribution in terms of theoretical framework and analysis, and our experiments comparison with multiple baselines.
>
> > **Q1**:  Why was the derivation of full scaling laws (e.g. following Cherti et al. 2023) was omitted in the work?
>
> **A**: Thank you for raising this concern. We would like to note that Cherti et al. (2023) focused on reproducing CLIP, which is empirical only. In contrast, we have rigorous theoretical analysis of the generalization error, which is not restricted to any model or data scale. Our experiments include the comparison with multiple baselines in the context of learning with reference models and our scaling law experiment serve to corroborate the presented theory.
>
> Following the reviewer's suggestion, we have added scaling law experiments for two more model scales (ViT-B/32 and ViT-L/14). The experiments have not completely finished since they take many days to run on large scales, but we can already observe that DRRho-CLIP has a better scaling trend across different model scales than OpenCLIP.
> - FLOPs vs. performance, without accounting for reference model cost: [anonymous link](https://github.com/icml2025drrhoclip/icml2025drrhoclip/blob/main/scaling_law_flops.pdf). The fitted scaling law for OpenCLIP is $E=5.77\cdot C^{-0.116}$, where $E$ is the ImageNet-1K Error and $C$ is the compute (GFLOPS). The fitted scaling law for DRRho-CLIP is $E=7.15\cdot C^{-0.127}$.
> - FLOPs vs. performance, accounting for reference model cost: [anonymous link](https://github.com/icml2025drrhoclip/icml2025drrhoclip/blob/main/scaling_law_flops_reference.pdf). The fitted scaling law for OpenCLIP is $E=5.77\cdot C^{-0.116}$, while for DRRho-CLIP it is $E=8.78\cdot C^{-0.135}$.
>
> > **Q2**: It would be also good to plot FLOPs vs. performance, accounting for FLOPs used by reference model.
>
> **A**: Links to the plots are provided in the answer to the above question. We want to highlight that the cost of leveraging a reference model can be amortized. In particular, since the reference model is frozen during training, we store the reference model features before training so that they can be reused multiple times across epochs and across different runs. Indeed, these features of the reference model have been already computed in the data filtering stage for creating existing datasets, e.g., LAION-2B (Schuhmann et al., 2022), DFN-2B (Fang et al., 2024) and Datacomp-1B (Gadre et al., 2023). In this case, we can directly leverage the features of the reference model without spending any resources computing them.

---

> > ### Comment · Reviewer_uJqj · 2025-04-03
> >
> > I am delighted to see further extension on scaling law study, and will update my score to 3.

---

> > > ### Author Response · Authors · 2025-04-07
> > >
> > > We thank the reviewer for their constructive feedback that helps improve our manuscript.

---

### Decision · Program_Chairs · 2025-05-01

**Decision:**

Accept (spotlight poster)

**Comment:**

This paper presents a rigorous theoretical foundation for Learning with a Reference Model (LEAR), a paradigm where a pre-trained model guides the training of a target model. By using LEAR with DRO, the authors offer a formal generalization analysis explaining this method's effectiveness. They further exploit the DRO-contrastive learning connection to develop DRRho-CLIP, for contrastive language-image pretraining with a reference model. Extensive experiments validate their theory and demonstrate superior performance over heuristic baselines.

The reviewers unanimously recommended acceptance of this paper, primarily due to the strength of its comprehensive theoretical and empirical contributions. I fully support this decision and believe the work will be of broad interest to both the ICML audience and the wider machine learning community.